# Structures of a non-ribosomal peptide synthetase condensation domain suggest the basis of substrate selectivity

Thierry Izoré[1,2,12 ✉], Y. T. Candace Ho[1,2,3,4,12], Joe A. Kaczmarski[5], Athina Gavriilidou[6], Ka Ho Chow[7], David L. Steer[1,8], Robert J. A. Goode[1,8], Ralf B. Schittenhelm[1,8], Julien Tailhades[1,2,3], Manuela Tosin[4], Gregory L. Challis[1,3,4,9], Elizabeth H. Krenske[7], Nadine Ziemert[10,11], Colin J. Jackson[3,5] & Max J. Cryle[1,2,3 ✉]

Non-ribosomal peptide synthetases are important enzymes for the assembly of complex peptide natural products. Within these multi-modular assembly lines, condensation domains perform the central function of chain assembly, typically by forming a peptide bond between two peptidyl carrier protein (PCP)-bound substrates. In this work, we report structural snapshots of a condensation domain in complex with an aminoacyl-PCP acceptor substrate. These structures allow the identification of a mechanism that controls access of acceptor substrates to the active site in condensation domains. The structures of this complex also allow us to demonstrate that condensation domain active sites do not contain a distinct pocket to select the side chain of the acceptor substrate during peptide assembly but that residues within the active site motif can instead serve to tune the selectivity of these central biosynthetic domains.

[1] Department of Biochemistry and Molecular Biology, The Monash Biomedicine Discovery Institute, Monash University, Clayton, VIC, Australia. [2] EMBL Australia, Monash University, Clayton, VIC, Australia. [3] ARC Centre of Excellence for Innovations in Peptide and Protein Science, Clayton, VIC, Australia. [4] Department of Chemistry, University of Warwick, Coventry, UK. [5] Research School of Chemistry, The Australian National University, Acton, ACT, Australia. [6] Interfaculty Institute of Microbiology and Infection Medicine Tübingen, Microbiology/Biotechnology, University of Tübingen, Tübingen, Germany. [7] School of Chemistry and Molecular Biosciences, The University of Queensland, St Lucia, QLD, Australia. [8] Monash Proteomics and Metabolomics Facility, Monash University, Clayton, VIC, Australia. [9] Warwick Integrative Synthetic Biology Centre, University of Warwick, Coventry, UK. [10] German Centre for Infection Research (DZIF), Partnersite Tübingen, Tübingen, Germany. [11] Interfaculty Institute for Biomedical Informatics (IBMI), University of Tübingen, Tübingen, Germany. [12] These authors contributed equally: Thierry Izoré, Y. T. Candace Ho. ✉email: thierry.izore@monash.edu; max.cryle@monash.edu

Non-ribosomal peptide synthetases (NRPSs) are important biosynthetic enzymes for the production of highly diverse and extensively modified peptides[1]. The diversity of non-ribosomal peptides is due to the combination of an ability to incorporate an expanded range of monomers compared to ribosomal peptide biosynthesis together with extensive modifications of the peptide both during and after chain assembly[2]. This is enabled by the modular architecture of NRPSs, which use repeating groups of catalytic domains to install one monomer into the growing peptide (Fig. 1a). Within a minimal chain extension module, an adenylation (A) domain performs the selection and activation of amino acid building blocks at the expense of ATP, prior to the loading of the monomer onto the phosphopantetheinyl (PPant) moiety of an adjacent peptidyl carrier protein (PCP) domain[1]. Chain assembly is then performed by condensation (C) domains, which typically accept two

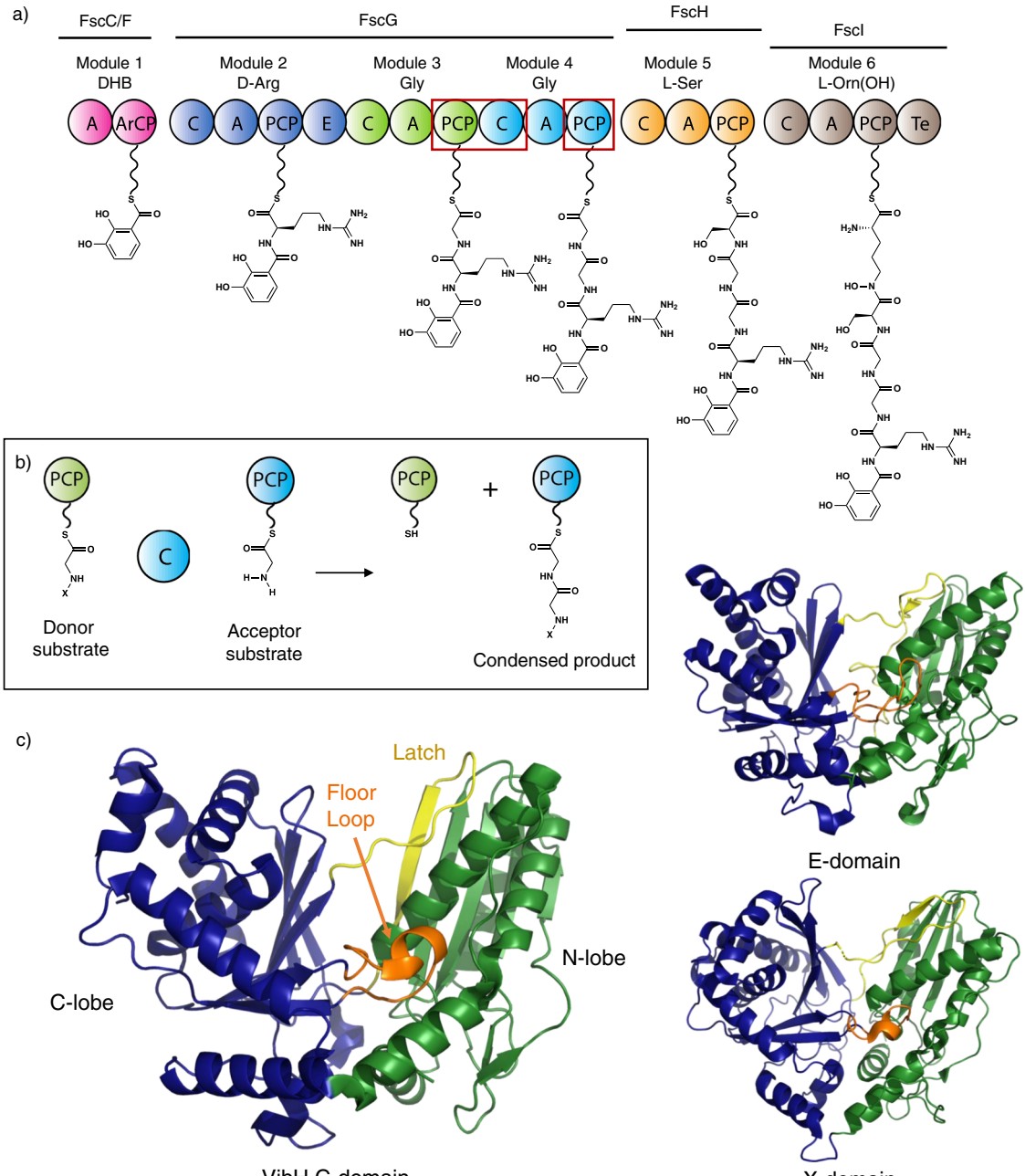

**Fig. 1 Non-ribosomal peptide biosynthesis and structures of C-type domains. a** Scheme representing the biosynthesis of a linear precursor of fuscachelin A; the domains structurally characterized in this manuscript are indicated by red boxes. **b** Condensation domains catalyze peptide bond formation most commonly between thioester intermediates bound to adjacent PCP domains; for mechanistic discussion see Supplementary Information. **c** Left: crystal structure of an archetypal C-domain (VibH from vibriobactin biosynthesis, PDB ID: 1A); Top right: crystal structure of an epimerization domain from tyrocidine biosynthesis (PDB ID: 2G); Bottom right: crystal structure of the cytochrome P450 recruitment (X)-domain from teicoplanin biosynthesis (PDB ID: 42). These domains are all comprised of a V-shaped pseudo-dimer of chloramphenicol acetyl transferase (CAT) domains (colored green and blue), with crossover regions including the latch (lemon yellow) and floor loop (orange). A - adenylation domain, DHB - 2,3-dihydroxybenzoic acid, ArCP - acyl carrier protein, C - condensation domain, E - epimerization domain, PCP - peptidyl carrier protein, Te - thioesterase domain, PPant moieties shown as undulated lines.

PCP-bound substrates and catalyze peptide bond formation through the attack of the downstream acceptor substrate upon the thioester of the upstream donor substrate (Fig. 1b)[3]. The first X-ray crystal structure of an NRPS C domain (VibH from the vibriobactin NRPS, Fig. 1c)[4] showed that they comprise a pseudo-dimer of the chloramphenicol acetyl transferase (CAT) enzyme fold, with key catalytic residues forming a conserved HHxxDG motif located at the interface between the two subdomains. In addition, it was shown that C domains harbor two catalytic tunnels that lead from the donor-PCP and acceptor-PCP domain binding sites to the active site and represent the access route for the donor and the acceptor substrates, respectively. This architecture has since been confirmed by other structures[4–16]. While the conserved central histidine (HHxxDG) is generally thought to act as the primary catalytic residue that promotes deprotonation of the α-amino group in the acceptor aminoacyl-PCP as it attacks the thioester, this remains a matter of debate[3]. Perhaps more importantly, the role C domains play in determining NRPS specificity is unclear, in part due to the lack of structural characterization of relevant PCP-bound C domain complexes.

Whilst the modular architecture of NRPSs has attracted great interest from the perspective of biosynthetic engineering[17–19], such efforts have not always been successful. This can be attributed to the complexity of the machinery combined with the necessity for non-native substrates to pass through multiple catalytic domains, each of which imparts a degree of specificity. A pertinent example of this is the recent recognition of the diverse functions of C domains in peptide biosynthesis, extending their well-established role in controlling peptide stereochemistry (working in concert with epimerization (E) domains) to gating in trans modifications, recruiting trans-acting enzymes and performing additional chemical transformations of their substrates during peptide bond formation (Fig. 1c)[20–24]. Whilst A domains are the main origin of structural diversity in non-ribosomal peptides[25], C domains play a key role in peptide bond formation and make important contributions to structural diversification in many valuable compound classes. Thus, gaining a deeper understanding of their function a high priority.

The structural analysis of key domains, complexes and complete modules has made major contributions to our understanding of how selectivity is achieved by NRPS assembly lines[26]. NRPS complexes are highly flexible, with domains connected by flexible linkers that allow the interactions between them to change during the process of chain assembly. However, the individual domains (and certain didomain complexes that represent metastable points along the catalytic pathway) are less dynamic and can be more readily studied by methods such as X-ray crystallography[7,13,27,28].

Structural characterization of key domain–domain complexes is thus an important goal to improve our understanding of NRPS selectivity. For example, structures of A domains in complex with PCP domains in distinct states, corresponding to substrate binding, substrate activation, and PPant loading have provided insight into the mechanisms underlying A domain selectivity[10,29]. However, C domains and C domain-containing complexes have proved more challenging to structurally characterize, with fewer examples reported to date (Fig. 1c)[3,26]. Furthermore, no structures of a C domain in complex with an acceptor PCP-domain bearing a substrate have been reported, which makes understanding the origins of C domain specificity for their acceptor substrates unclear, and also limits our understanding of the role of active site residues in C domain catalysis[3].

To address this, we report the structure and biochemical characterization of complexes of a PCP domain bearing a stable analog of the acyl acceptor complexed to the acceptor site of a C domain from the NRPS that biosynthesizes fuscachelin in the thermophile *Thermobifida fusca* (Fig. 1a)[30]. This structure reveals that the interface between the PCP and C domains is dominated by hydrophobic interactions and that access to the C domain active site is gated by an arginine residue that prevents unloaded PCP-substrates from accessing the active site of the C domain. The C domain is shown to be tolerant of a small range of aliphatic amino acid acceptor substrates, with the limited acceptance of other substrates rationalized through interactions with key residues within the C domain active site. We demonstrate that C domains do not appear to contain an "A domain-like" side chain selectivity pocket to control their acceptor substrates and resolve how substrates engage with central catalytic residues in C domains, both of which are key unanswered questions central to NRPS-mediated peptide biosynthesis.

## Results

**Structure of the PCP$_2$-C$_3$ didomain.** To elucidate the structure of a C domain with a PCP domain bound in the acceptor site, we screened several systems including a thermophilic example of a PCP$_2$-C$_3$ didomain (containing the second PCP and the third C domain) of the fuscachelin NRPS from the thermophilic organism *Thermobifida fusca* (Fig. 1a [red rectangle])[30]. Expression of the fuscachelin PCP$_2$-C$_3$ didomain in *E. coli* yielded 0.8 mg/L of culture of stable protein and afforded crystals that grew rapidly in 18–22% w/v PEG 3350 and 0.17-0.3 M magnesium formate at room temperature. Crystals were harvested, cryoprotected in 20–30% glycerol and diffraction data collected at the Australian Synchrotron, with initial phases obtained from a single-wavelength anomalous diffraction experiment (SAD) using xenon-derivatized crystals (see Methods section). The crystals belonged to the P2$_1$2$_1$2$_1$ space group, with the unit cell comprising two highly similar copies of the PCP$_2$-C$_3$ construct (RMSD (all atoms) 0.74 Å).

The PCP$_2$-C$_3$ didomain structure we obtained from these experiments was solved at a resolution of 2.2 Å (PDB ID 7KVW; Fig. 2a and Supplementary Table 1). When considered separately, the overall folds of both the PCP$_2$ domain and C$_3$ domain were consistent with previously reported structures[26]. The PCP$_2$ domain comprises a 4-helix bundle with a small α-turn between helices 1 and 2 (seen in most crystal structures but absent from NMR structures); the serine residue that is the site of 4′-phosphopantetheine (PPant) attachment is located at the start of helix 2 (Fig. 2b). Of the published crystal structures of PCP domains, this structure is most similar to the PCP domain found in the PCP-Te/R didomain NRPS construct from the archaeon *Methanobrevibacter ruminantium* M1 (PDB ID 6VTJ; RMSD (all atoms) 1.3 Å, 32% sequence similarity, see Supplementary Table 2). The C$_3$ domain of the didomain resembles other members of its class (see Supplementary Table 3), comprising a pseudo-dimer of CAT domains with bridge (R2923 to T2944) and floor loop (A2843 to L2858) regions (Figs. 1c and 2c). The catalytic residues sit at the core of the C$_3$ domain and can be accessed from the bulk solvent via tunnels formed along the interface of the two pseudo-domains (Fig. 2d). Differences in the relative position of these two halves are observed in structures of C domain homologs and can alter the size and character of the acceptor and donor catalytic tunnels[3]. A superimposition of the fuscachelin C$_3$ domain with two well-characterized C domains (from surfactin and linear gramicidin NRPSs)[7,12] highlights this, with a pronounced difference in displacements observed when comparing the fuscachelin C$_3$ domain and Srf-A domain (Supplementary Fig. 1). This aspect of C-domain conformational flexibility and diversity is currently not

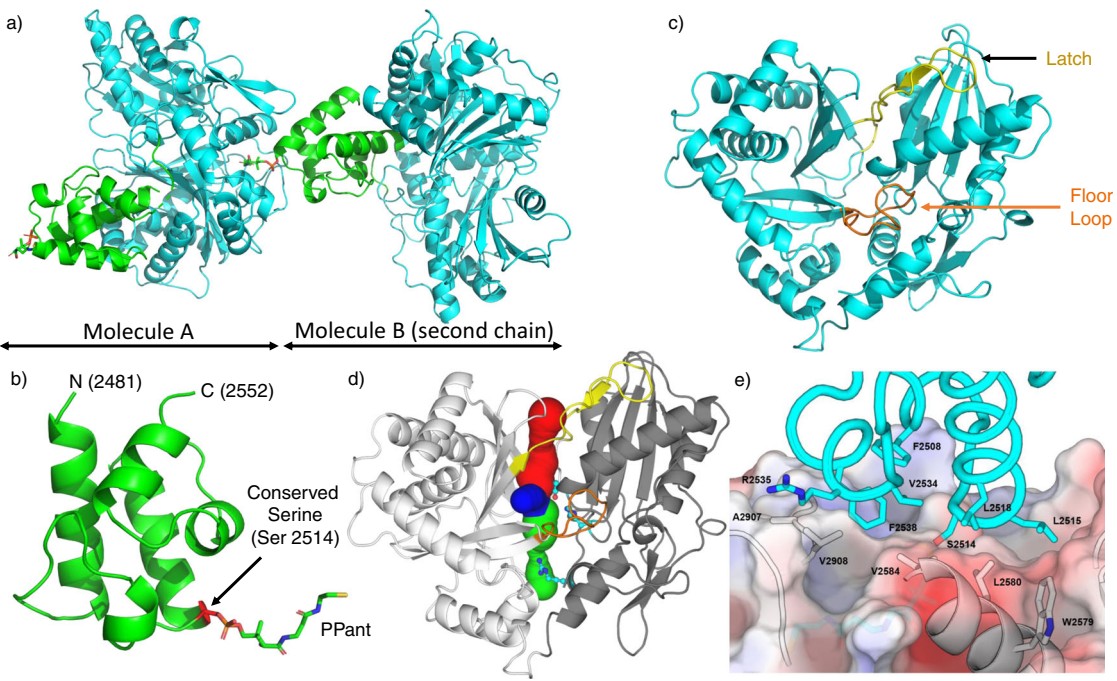

**Fig. 2 Overview of the structure of the PCP₂-C₃ didomain from fuscachelin biosynthesis. a** Crystal structure of the PCP₂-C₃ didomain (PDB ID 7KVW) showing two chains, with the PCP domain positioned at the acceptor site of the C domain from another molecule (C domain shown in cyan, PCP shown in green). **b** Structure of the PCP₂ domain, a 4-helix bundle with an additional small α-turn between helices 1 and 2 with the PPant arm bound to Ser2514. **c** Structure of the C₃ domain, displaying a pseudo-dimer of CAT domains (latch and floor loop regions represented in yellow and orange, respectively); the donor binding site is at the top of the figure and the binding acceptor site is at the bottom of the figure. **d** C₃ domain showing the donor tunnel (blue), acceptor tunnel (green), and a third tunnel (red) converging on the active site (blue). The tunnel lining residue R2577 and the active site residues E2702 and H2697 are shown as cyan sticks. **e** The hydrophobic interface between the PCP₂ domain (cyan sticks and ribbon) and C₃ domain (surface representation + gray sticks and ribbon). N - N-terminal, C - C-terminal, PPant - phosphopantetheinyl.

broadly understood, although recent efforts have been made to understand these conformational differences in terms of the accessibility of the substrates to the active site the C-domain[9].

In the PCP₂-C₃ didomain structure, the PCP₂ domain sits at the acceptor-PCP binding site (near the opening of the acceptor substrate channel) on the C₃ domain from the second chain in the asymmetric unit. The interface between the PCP₂ domain and C₃ domain is mostly hydrophobic in nature (537/510 Å² buried surface area (chain A/B) excluding PPant), with the side chains of V2534, L2515, L2518, F2508, and F2538 of the PCP domain playing a major role in the interaction along with residues A2907, V2908, V2584, L2580, and W2579 of the C domain (Fig. 2e and Supplementary Tables 4 and 5). This interface is reminiscent of the hydrophobic interaction pattern described in other structures of PCP domains found docked at the acceptor site of C domains (SrfA-C (PDB ID 2VSQ)[12], AB3403 (PDB ID 4ZXH)[10]; see also Izoré et al.[26]). These interfaces center around a hydrophobic residue (L2515) immediately following the serine to which the PPant is attached (S2514) and at least one hydrophobic residue ~20 amino-acids after the serine residue. R2906 also plays an important role in positioning the PCP domain via interactions with the phosphate moiety of the PPant arm. In the PCP₂-C₃ structure, these residues are V2534 and the aliphatic moiety of R2535 that interacts with V2908 of the C domain. The overall orientation of the PCP domain relative to the C domain is similar to what has been observed in the structures of SrfA-C[12] and ObiF1 (PDB ID 6N8E)[8] (Supplementary Fig. 2A, B), whilst other structures contain a PCP domain that is rotated by several degrees around the conserved serine (AB3403[10], LgrA (PDB ID 6MFZ)[7]; Supplementary Fig. 2C, D). Although the overall orientation of these PCP domains in relation to the C domain are different, it is important to note that the position of the

PPant-modified serine (located at the beginning of the second helix) is always maintained at the entrance of the acceptor substrate channel of the C domain.

Since the PCP₂ domain precedes the C₃ domain in the fuscachelin NRPS, we had expected that the PCP₂ domain would be positioned at the donor-PCP binding site of the C₃ domain. We were surprised, therefore, to find that this construct crystallized with the PCP₂ domain positioned at the acceptor-PCP binding site of the C₃ domain of the second chain in the asymmetric unit (Fig. 2a). Given that the PCP₂ and PCP₃ domains of the fuscachelin NRPS are highly similar (65% sequence identity, Fig. 3), and that PCP domains can act as both aminoacyl donors and acceptors for C domains, we rationalized that the arrangement observed in our structure is a valid model of an acceptor-PCP-bound C domain. Indeed, when we determined the structure of the isolated PCP₃ domain, we found its structure (PDB ID 7KW3) to be highly similar to the PCP₂ domain (RMSD (all atoms) 2 Å; Fig. 3a–c). Importantly, the residues at the interface with the C domain are conserved or highly similar (Fig. 3d). Furthermore, computational docking of the PCP₃ domain onto the acceptor-PCP binding site of the C₃ domain showed that it binds in an almost identical orientation to the PCP₂ domain in the structure of the PCP₂-C₃ didomain (Supplementary Fig. 4). This supports the notion that the PCP₂-C₃ didomain structure is a valid representation of an acceptor-PCP-bound C-domain.

Analysis of the PCP₂-C₃ didomain structure (PDB ID 7KVW) revealed extra density extending from the conserved Ser (S2514) at the beginning of helix 2 of the PCP domain. This serine residue is the target of phosphopantetheinyl transferases, a class of enzymes that attach the essential PPant moiety to PCP domains. Mass spectrometric analysis of the PCP₂-C₃ didomain construct

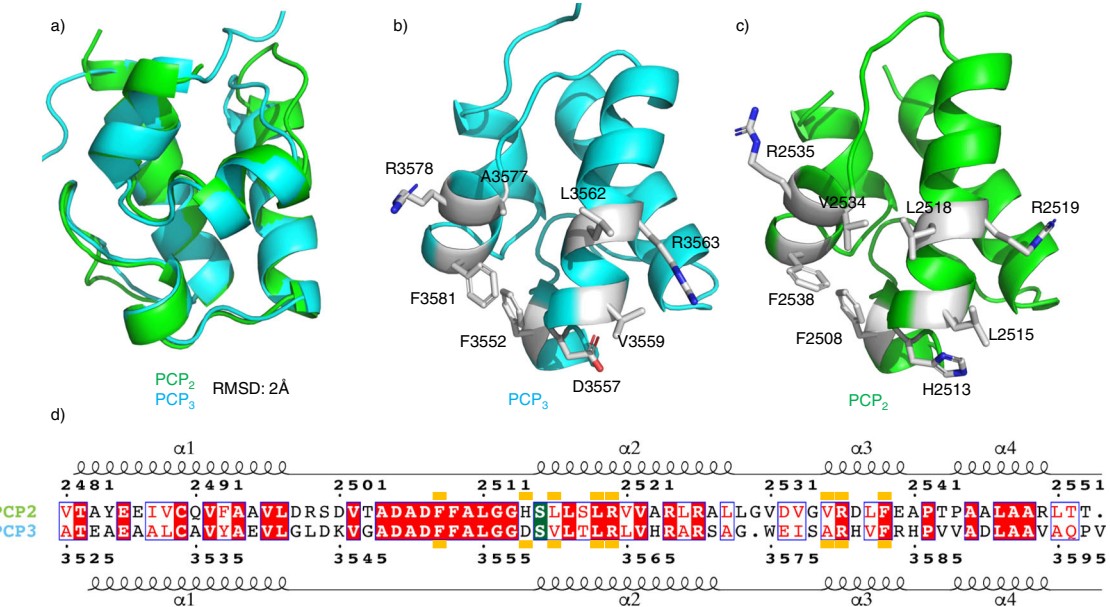

**Fig. 3 Comparison of PCP$_2$ and PCP$_3$ domains from fuscachelin biosynthesis. a** Structural alignment of PCP$_2$ and PCP$_3$ domains (RMSD 2 Å). **b** Crystal structure of the PCP$_3$ domain (PDB ID 7KW3) showing the position of side chains for interaction with C-domain based on PCP$_2$. **c** Crystal structure of the PCP$_2$ domain showing side chains interacting with the C-domain. **d** Sequence alignment of both PCP$_2$ and PCP$_3$ domains with the C domain interface indicated by orange blocks (conserved residues highlighted in red, similar residues shown in red text); site of posttranslational modification highlighted in green.

revealed a 340 Dalton mass increase, consistent with attachment of PPant to S2514, likely installed by the phosphopantetheinyl transferase EntD that phosphopantetheinylates some PCP domains when they are expressed in *E. coli*. Indeed, expression of the PCP$_2$-C$_3$ didomain construct in an *entD* mutant[31] showed no increase in mass, supporting this hypothesis. Having confirmed the presence of a PPant arm, we modeled this into the electron density observed in our structure. Interestingly, we found that this did not extend into the active site of the C domain, but instead curled back towards the outer surface of the C domain (Fig. 4a). The side chain of R2577 appears to block the channel that leads to the active site of the C domain (Fig. 4a). Molecular dynamics simulations initiated from structures of the C$_3$ domain (with the PCP-PPant removed) highlight the intrinsically dynamic nature of the acceptor substrate channel and the important role that R2577 has in modulating its shape and size (Supplementary Fig. 5). This residue forms the bottleneck of the channel and samples alternate rotamers (primarily rotation around chi-3) that, in concert with a displacement of alpha-helix 1, largely determines its size. When we compared our PCP$_2$-C$_3$ didomain structure with published structures of other C domains in complex with a PPant-modified PCP domain, we found residues with shorter side chains at this position (G21 in AB3403[10] and A18 in ObiF1[8]), resulting in channels that do not block PPant access. Next, we identified all available C-domains from the MiBiG database and computed multiple sequence alignments ($^LC_L$ and $^DC_L$ sequences; Superscript indicates the stereochemistry of the C-terminal residue of the donor substrate, subscript indicates the stereochemistry of the acceptor substrate) in order to discern the typical amino acid found at this position. Interestingly, this Arg residue appears largely conserved in $^LC_L$ domains (73% harbor an Arg at this position), but is not seen in $^DC_L$ domains (Gly (80%) or Ala (4%) are found instead (Supplementary Fig. 6)). Whilst it was unclear what role this residue plays in NRPS function, we hypothesized that it could influence access to acceptor channel of the C domain.

**Effect of R2577G mutation on substrate position**. To verify the role of the R2577 in controlling access to the catalytic channel, we generated the Arg to Gly mutant (R2577G) of the C$_3$ domain. To control the modification state of the PCP$_2$ domain, the mutant PCP$_2$-C$_3$ didomain construct was expressed in the *entD* mutant of *E. coli*[31]. After purification, the protein was modified using the promiscuous PPant transferase Sfp R4-4 mutant[32] and coenzyme A (CoA; see Methods section) to ensure homogeneous PPant loading. Similar to the wild-type construct, the protein expressed well and crystallized in the same conditions. Crystals diffracted to 2 Å and the structure was phased using molecular replacement with the previous model (PDB ID 7KW2; Supplementary Table 1). The structure of the R2577G mutant is very similar to that of the wild-type protein, with the PCP$_2$ domain sitting at the acceptor site of the C$_3$ domain (RMSD (all atoms) 1.2 Å compared to wild type). The first noticeable difference is a small rotation of the PCP domain in relation to the C domain and slight alterations in the PCP interacting regions of the C domain, likely attributable to the R2577G mutation allowing the first helix of the C domain to sit deeper in the acceptor channel (Supplementary Fig. 7)[5]. The major difference, however, is the positioning of the PPant moiety, which now fully extends thought the acceptor channel into the active site (Fig. 4b and Supplementary Fig. 8) in a similar way to that seen in the ObiF1, SrfA-C, and AB3403 structures[8,10,12]. This observation supports the hypothesis that R2577 acts to control substrate access to the active site of the C domain. One possibility is that this process operates by charge repulsion: when an aminoacyl-PPant approaches the acceptor channel, the ammonium group of the substrate triggers the rotation of the Arg side chain due to charge repulsion, which opens the channel, allowing the aminoacyl-PPant to enter it. This would explain our inability to crystallize the wild-type PCP$_2$-C$_3$ construct loaded with PPant derivatives lacking an amino group (such as propionyl and propan-1,3-dioyl[33]), due to interactions that interfere with crystallization when the substrate is not bound in the acceptor channel of the C domain. To further explore this mechanism, we next turned to the characterization of the

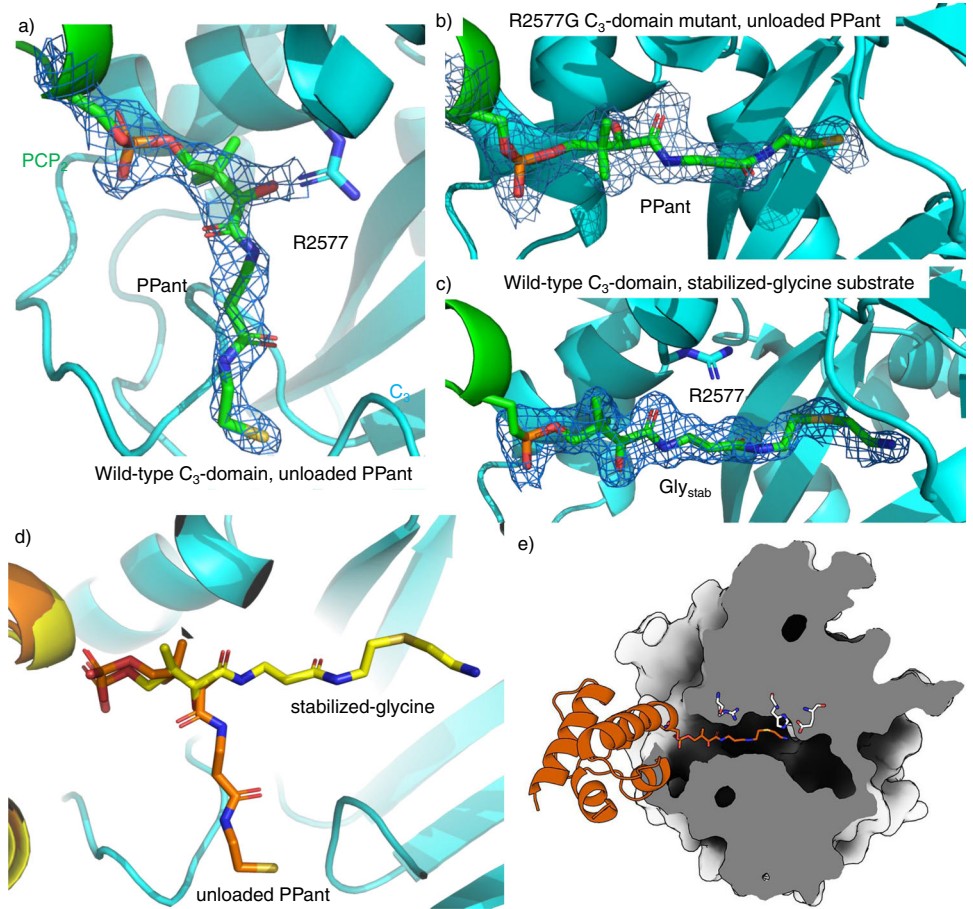

**Fig. 4 PCP₂-C₃ interaction interfaces showing the differences in substrate acceptance. a** Structure of WT C₃ domain with unloaded PPant (7KVW), showing the PPant not extending into the C₃-domain as the side chain of R2577 prevents the PPant accessing the C₃-domain active site. **b** Structure of R2577G C₃ domain with an unloaded PPant (7KW2), showing the PPant fully extended into the C₃-domain catalytic channel. **c** Structure of WT C₃ domain where the PPant is loaded with a Gly$_{stab}$ substrate (7KW0), rotated 90° anticlockwise compared to panels (**a**) and (**b**). Here, PPant-Gly$_{stab}$ extends fully into the catalytic channel. **d** Comparison of the positioning of the unloaded PPant (orange) and PPant-Gly$_{stab}$ (yellow) within the C₃ domain. **e** Cutaway representation of the C₃ domain indicating the path of the PPant-Gly$_{stab}$ substrate from the PCP₂ domain (shown in orange). All densities shown as 2Fo-Fc maps, contoured at 1σ and using a carve value of 1.8 Å.

PCP₂-C₃ construct with an aminoacyl group appended to the PPant thiol group.

**Structure of the amino acid acceptor bound substrate.** To append the glycyl substrate of module 3 to the PCP₂ domain, we attempted to load the apo-PCP₂C₃ didomain using Sfp and the CoA thioester of glycine. Crystals in the same space group were readily obtained using the same method as for the two previously described structures. Somewhat surprisingly, in this structure it was clear that the electron density corresponding to the PPant did not sit in the acceptor channel but rather followed the same path as the substrate-free PPant, appearing to be repelled by R2577. However, upon refinement it became clear that the glycyl thioester had been hydrolyzed during crystallization. This forced us to explore alternatives to thioester-tethered amino acids, and we chose to use an analog of the aminoacyl-CoA with a thioether, hence removing the reactive carbonyl that makes the thioester susceptible to nucleophilic attack. This results in a non-hydrolyzable substrate analog that is still tethered to the PPant via a C–S bond and has a very similar structure to the real substrate (Supplementary Fig. 9), circumventing issues encountered with other stabilization strategies[34]. To obtain crystals of the PCP₂-C₃ construct with this substrate analog (hereafter referred

to as Gly$_{stab}$) bound, we again used Sfp to attach PPant-Gly$_{stab}$ to the PCP domain. This construct was then crystallized as previously, resulting in diffraction to a resolution of 1.9 Å (PDB ID 7KW0; Supplementary Table 1).

The overall structure of the Gly$_{stab}$-loaded PCP₂-C₃ construct was highly similar to the holo-PCP₂C₃ construct (572/532 Å² buried surface area (chain A/B) excluding PPant). In the Gly$_{stab}$ structure, however, the density for the PPant extends through the acceptor channel of the C domain into the active site, as observed in the structure of the R2577G mutant (Fig. 4c, d). R2577 now forms weak interactions with two of the carbonyl oxygen atoms in the PPant arm (3.7 Å and 3.8 Å), possibly acting as a ratchet to hold the PPant arm (and substrate) in the correct position until after peptide bond formation has occurred (Supplementary Fig. 10). Analysis of the residues found in the PPant channel also found a similar trend of conservation as was the case for R2577, in which $^{L}C_L$ and $^{D}C_L$ domains show different patterns of conservation (Supplementary Fig. 11). The PPant-Gly$_{stab}$ extends completely into the active site (Fig. 5a), with the terminal amine of Gly$_{stab}$ stabilized by hydrogen-bond interactions (Fig. 5b). Of particular interest, given the lack of clarity over the role of the active site histidine in the HHxxDE motif, is its close proximity (3.6 Å) to the amino group of the Gly$_{stab}$ moiety. An ordered water molecule also sits close (2.9 Å) to this amino group, where

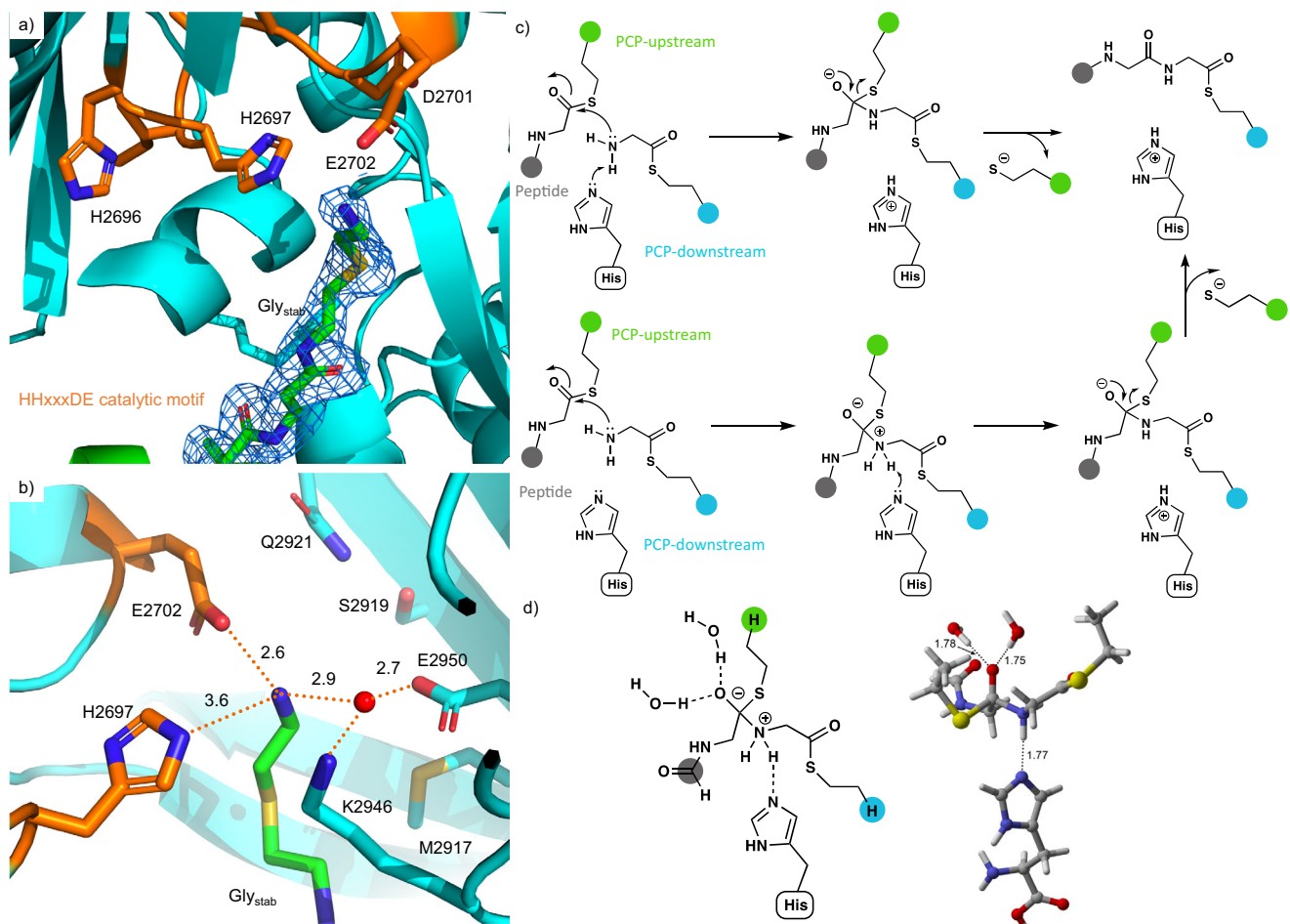

**Fig. 5 The C₃ domain catalytic site showing the position of PPant-Gly_stab. a** PPant-Gly_stab substrate extends fully into the active site, approaching the active site HHxxxDE motif (H2696 to E2702); electron density shown as a 2Fo-Fc map (PDB ID 7KW0). **b** The Gly_stab substrate is stabilized by a network of hydrophilic interactions. Note that residues M2917, S2919, Q2921, P2941, and E2950 are in a position that could potentially interact with the side chain of alternate acceptor substrates. **c** Mechanism of peptide bond formation via concerted N–C bond formation and N-deprotonation (upper pathway) or sequential N–C bond formation and N-deprotonation (lower pathway); donor PCP shown in green, acceptor PCP shown in cyan, peptide is shown in gray. **d** Zwitterionic intermediate in the sequential N–C bond formation/N-deprotonation pathway, in which the oxyanion is stabilized by two water molecules and the ammonium ion forms a hydrogen bond to histidine (see Source Data).

it likely forms a hydrogen bond. In order to determine whether the intrinsic mechanistic preference of the amide bond-forming reaction is stepwise or concerted, we calculated the reaction of a model donor, acceptor, and imidazole base in solution with density functional theory (Fig. 5c, see Supplementary Discussion for details of the mechanistic investigation). The attack of the model amine on the thioester strongly prefers a stepwise mechanism in which N–C bond formation precedes N deprotonation by the imidazole, rather than a concerted mechanism in which these two events take place simultaneously. Therefore, we predict that the enzyme-catalyzed amide bond formation likely involves a similar sequence, with a distinct zwitterionic (oxyanion/ammonium) intermediate (Fig. 5d). A distinct energy barrier is observed for proton transfer from the zwitterionic intermediate to the imidazole group of the active site histidine residue. This may explain why the mutation of this central histidine residue does not completely abolish activity in some C domains, as an active site water molecule could instead play the role of an alternate base[11]. The calculations show that the formation of at least one hydrogen bond to the oxyanion is key to stabilizing the zwitterionic intermediate. We also observed the close interaction of the atypical E residue in the HHxxxDE motif (which is typically a Gly in most C-domains) with the nitrogen

atom of Gly_stab (2.6 Å). It is important to note that Gly_stab sits in a different position to the aminoacyl mimic in a previous model of a C domain bound to the acceptor substrate – in these structures the aminoacyl mimic does not enter into the active site as far as observed in our Gly_Stab-PCP₂-C₃ complex (Supplementary Fig. 12)[11].

**Exploring C domain activity and specificity of the PCP₂-C₃ construct**. To test the activity and selectivity of the C domain, as well as the effect of mutating key residues, we first needed to generate an activity assay for the C domain using the PCP₂-C₃ construct and downstream PCP₃ domain. Given that the interaction between PCP and C domains is weak and transient in nature[26], we first validated the importance of this restraint in an assay using separately isolated PCP₂-C₃ (loaded with a synthetic dihydroxybenzoic acid (DHB)-D-Arg-Gly donor substrate) and PCP₃-Gly constructs. This experiment revealed no elongation when these constructs were incubated together. Thus, we turned to the use of a fused PCP₂-C₃-PCP₃ construct, albeit one in which the PCP-constructs could be separately loaded with substrates prior to generation of the fused complex (Fig. 6a). To accomplish this, we cloned the donor PCP₂-C₃ construct with a C-terminal

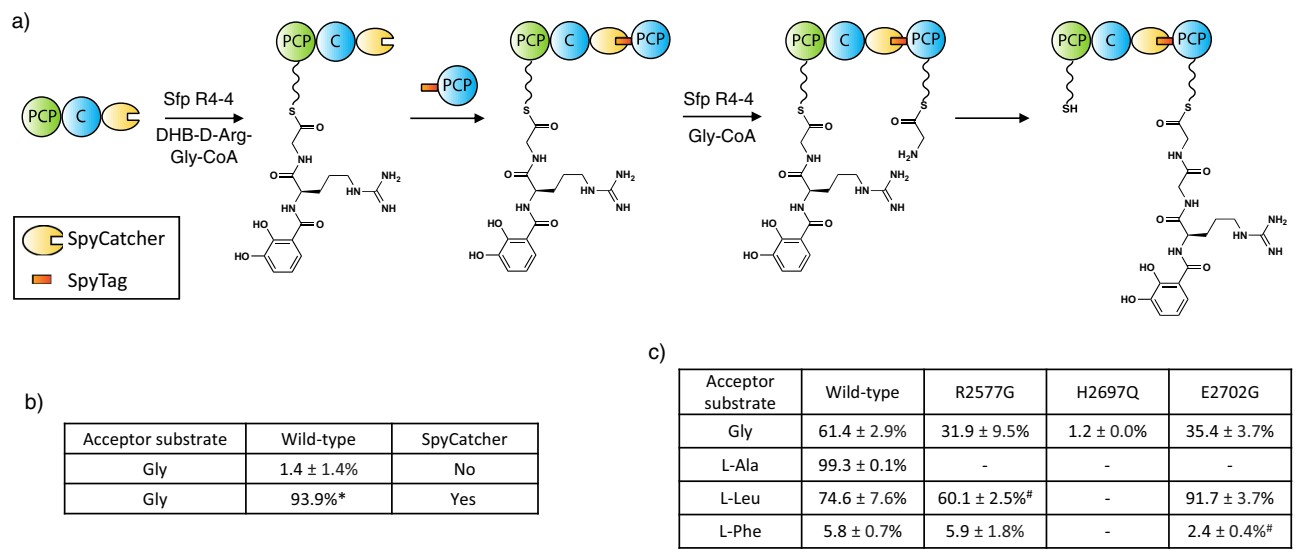

**Fig. 6 C₃-domain condensation assays. a** Scheme of the condensation reaction using PCP₂C₃ SpyCatcher and Spytag-PCP₃ constructs. **b** Level of tetrapeptide formation demonstrated by the WT C₃-domain with or without SpyCatcher and SpyTag; the reaction was performed using a DHB-ᴅ-Arg-Gly donor substrate and a Gly acceptor substrate. **c** Level of tetrapeptide formation by WT and different C₃-domain mutants using BA-ᴅ-Arg-Gly as a donor substrate and different aminoacyl acceptor substrates. All reactions performed in triplicate, unless specifically stated (* single reaction; # duplicate); see Supplementary Figs. 30–35 for traces; for data see [http://proteomecentral.proteomexchange.org/cgi/GetDataset?ID=PXD024004].

**b)**

| Acceptor substrate | Wild-type | SpyCatcher |
|---|---|---|
| Gly | 1.4 ± 1.4% | No |
| Gly | 93.9%* | Yes |

**c)**

| Acceptor substrate | Wild-type | R2577G | H2697Q | E2702G |
|---|---|---|---|---|
| Gly | 61.4 ± 2.9% | 31.9 ± 9.5% | 1.2 ± 0.0% | 35.4 ± 3.7% |
| L-Ala | 99.3 ± 0.1% | - | - | - |
| L-Leu | 74.6 ± 7.6% | 60.1 ± 2.5%# | - | 91.7 ± 3.7% |
| L-Phe | 5.8 ± 0.7% | 5.9 ± 1.8% | - | 2.4 ± 0.4%# |

SpyCatcher domain and the acceptor PCP₃ with an N-terminal SpyTag peptide[35]. The SpyCatcher/SpyTag system are based on an engineered CnaB2 domain of the *Streptococcus pyrogenes* FbaB protein, from which the C-terminal strand (SpyTag, 13 amino acids) has been separated from the rest of the domain (SpyCatcher, 12.3 kDa). Isopeptide bond formation involving residues on both SpyCatcher/SpyTag components then results in the covalent bonding of the two fragments[35]. This system allows for the separate loading of the substrates on the PCP domain of each construct using Sfp and synthetic CoA substrates whilst also allowing the reconstitution of the NRPS assembly line.

Using this experimental setup, we confirmed that the condensation reaction was performed as expected, with high levels of conversion of the canonical donor DHB-ᴅ-Arg-Gly tripeptide into the Gly-extended tetrapeptide, as determined by high-resolution LC-MS/MS experiments (Fig. 6b, see Supplementary Figs. 30–35). Next, we tested a simplified benzoic acid (BA)-ᴅ-Arg-Gly donor substrate in these assays, which showed acceptable levels of conversion (61%) and hence we retained this simplified substrate for all subsequent assays. With a functional condensation assay in hand, we first could verify that the stabilized Gly_stab acceptor substrate was a functional mimic of Gly in this C domain (Supplementary Fig. 13) using intact protein MS together with PPant ejection (see Methods section). With confidence that the Gly_stab structure represents a functional acceptor substrate-bound C domain state, we then set out to investigate the effect that mutating key residues had on the condensation activity. Firstly, we confirmed that the R2577G mutant C domain retained activity (with Gly), although this was reduced compared to the wild-type C domain (32%), possibly due to the loss of stabilizing interactions with the PPant arm (Fig. 6c and Supplementary Fig. 10). We next generated an active site H2697Q mutant and determined that H2697 is indeed essential for activity with this C domain, as the mutant only retains ~1% of the WT activity with Gly as the acceptor substrate (Fig. 6c).

In addition to Gly and Gly_stab, we found that C₃ could also accept PPant-linked ʟ-Ala and ʟ-Leu as substrates, with 99% and 75% conversion levels, respectively (Fig. 6c). In contrast,

PPant-linked ʟ-Phe was a poor substrate, with minimal (6%) levels of conversion. In order to rationalize these differences, we analyzed the structures and performed molecular docking of these alternate substrates (ʟ-Ala, ʟ-Leu, and ʟ-Phe) into the structure of the C₃-domain. First, assuming that the position of Gly_stab in our Gly_stab-PCP₂-C₃ complex represents that catalytically competent conformation, and that alternate amino acid acceptor substrates must bind in a way that positions the terminal amine group in a similar position, we identified several residues in the central cavity that would likely interact with the side chain of an alternate acceptor substrate. In particular, the side chains of M2917, S2919, Q2921, P2941, and E2950 could contribute a putative side chain binding pocket for this C₃-domain, in a manner reminiscent of A-domains (Fig. 5b). Computational docking of alternative substrates into the C₃-domain revealed that side chains of ʟ-Ala and ʟ-Leu could be accommodated by the active site cavity's side-chain binding pocket and had top scoring poses that positioned the terminal amine towards the catalytic residues (although the ʟ-Leu pose was slightly strained, Supplementary Fig. 14). In contrast, the bulky side chain of ʟ-Phe could only be accommodated within the central cavity in poses that positioned the terminal amino acid amine away from the catalytic histidine and that would not be compatible with catalysis (Supplementary Fig. 14). In order to discover possible correlation between putative pocket residues and the reported activity of the downstream A-domain we used the carefully curated, non-redundant MiBiG dataset by extracting all C-A linker regions (see Methods section) with known acceptor domain specificity and computing a multiple sequence alignment to identify the residues of interest. Analysis of these residues (M2917, S2919, Q2921, P2941, and E2950) compared to the reported activity of the downstream A-domain did not reveal any correlation between acceptor substrate and these possible "pocket" residues (Spearman's rho: −0.05), indicating the lack of a C-domain side chain binding pocket and hence there being no "C domain code" comparable to those found with A domains. This result was supported by a principal component analysis that also showed no patterns of correlation (Supplementary Fig. 15)[25,36]. Our results

do however indicate that alterations in the C domain active site can lead to changes in selectivity, and hence we turned to further analysis of the residues within the active site motif.

Although most C domains contain a canonical HHxxxDG motif[37], the $C_3$ domain from the fuscachelin NRPS features an unusual HHxxxDE variant. We hypothesized that, in absence of a side chain in the acceptor substrate (Gly) to position the acceptor substrate, the role of this glutamate (E2702) could be to stabilize and orient the acceptor substrate amine group to ensure an efficient nucleophilic attack of the donor substrate thioester. To observe how this motif is connected to acceptor substrate size, we extracted all C-domain sequences with known downstream A-domain specificity from the MiBiG database. Indeed, an analysis of these C domains demonstrated that there is a higher proportion of modified motifs where the acceptor substrate is small as opposed to traditional HHxxxDG containing C domains (Supplementary Fig. 16). To test this hypothesis, we mutated this glutamate to its canonical glycine residue (E2702G) and performed condensation reactions with Gly as the acceptor substrate. As expected, the condensation level with Gly as the acceptor substrate was reduced by almost half (61% to 35%) when compared to the WT, demonstrating the non-essential, although beneficial role of this glutamate residue. Interestingly, while the E2702G mutation had reduced activity with the Gly acceptor substrate, this substitution improved the activity for PPant-linked L-Leu from 75% to 92% (Fig. 6c). This result indicates that the E2702 residue can play a particularly important role in supporting condensation reactions involving Gly as an acceptor substrate, but may be detrimental for other acceptor substrates. Computational docking of Gly-PPant and $Gly_{stab}$-PPant into a model of the E2702G $C_3$ mutant reveals how the removal of the glutamic acid results in substrate poses that are unlikely to be compatible with catalysis, with the terminal amine of the substrates instead interacting with Glu2950 (Supplementary Fig. 17).

## Discussion

Non-ribosomal peptide synthetases are widely recognized for their impressive selectivity in assembling specific peptide products. While the role of the A domain in substrate selection is clear, the possible role of C domains as a second selectivity filter during peptide assembly has been less well defined. Early studies suggested C domains may show selectivity towards their acceptor substrates[36], but more recent work has questioned this[25].

The structural characterization and bioinformatics analysis we have performed of PCP-bound acceptor complexes in this work shows no general correlation between the size or chemical nature of the acceptor amino acid side chain and potential side chain binding residues in the C domain. Whilst C domain selectivity has recently been characterized in glycopeptide antibiotic biosynthesis[21], there the mechanism rather acts to ensure that important modifications of the PCP-bound aminoacyl thioester are performed prior to condensation. Whilst some selectivity for the amino acid substrate is seen here, for example, in the low conversion (albeit still present) of L-Phe, this appears likely to be due to the significant difference between the small, flexible Gly substrate and L-Phe, with its large, rigid side chain. The influence of the atypical HHxxxDE motif of this C-domain is also seen on lower levels of acceptance of larger amino acids (such as Leu), which can be released upon conversion of the motif into the typical HHxxxDG sequence. This demonstrates the versatile nature of C domains for tolerating active site modifications, some of which can play important additional roles in supporting catalysis[24].

Within the active site, the amino group of the aminoacyl acceptor lies close to the central histidine residue, with calculations suggesting that this residue could indeed act as a base to deprotonate the zwitterionic intermediate. Further characterization of the PCP-C complex shows that the PCP binding site of the C domain is, as anticipated, dominated by hydrophobic interactions and is one that is relatively flexible with regards to the PCP domain[26]. Access of the PPant arm to the C domain active site appears to be gated by R2577, which repels the unmodified PPant arm (or neutral/negatively charged substrates) in favor of the aminoacyl-PPant. Whilst this residue is largely conserved in $^LC_L$ domains, it is typically Gly or other small residues in $^DC_L$ domains, which we have confirmed allows the unmodified PPant into the C-domain active site. One hypothesis for the role of this residue would be to prevent the unwanted "pass-through" of donor substrates without elongation (e.g. from $PCP_2$ to $PCP_3$). Examples of NRPS-dependent pathways in which CP-bound substrate transfer could occur reveals that the C domains implicated bear the Arg to (Gly/small) amino acid mutation (e.g. burkholdac biosynthesis)[38], which provides some support for this hypothesis. For $^DC_L$ domains, mutation of this Arg residue could be a requirement due to the need for E domain-catalyzed inversion of stereochemistry prior to chain elongation, as we note that the Arg to (Gly/small) mutation generally appears to be somewhat deleterious to peptide conversion levels, possibly due to a lack of interactions between the Arg and PPant arm in these C-domains. We anticipate that the structural snapshots presented here will pave the way for studies to probe the roles of this Arg residue as well as other active site residues in C domain catalysis, which is important due to the ever-increasing roles of C-type domains in non-ribosomal peptide biosynthesis.

## Methods

**$PCP_2$-$C_3$ and $PCP_3$ constructs**. Gene fragments encoding the desired regions of FscG (UniProt ID Q47NR9) were amplified by PCR from *Thermobifida fusca (ATCC 27730)* genomic DNA using primers #1 and #2 for $PCP_2$-$C_3$ and #3 and #4 for $PCP_3$ (Supplementary Table 6). Target vectors (pOPIN-S and pET28a, respectively) were linearized using primers #15 + #16 (for pOPINS-S) and #13 + #14 (for pET28a). Amplicons were analyzed on a 0.8% agarose gel in TBE buffer and the DNA subsequently gel-extracted and purified using the GeneJET Gel Extraction Kit (Thermo Fisher Scientific). The extracted PCR products were then used in an In-Fusion® cloning reaction as per the manufacturer's instructions ($PCP_2$-$C_3$ cloned into pOPINS-S and $PCP_3$ cloned into pET28a). In-Fusion® cloning reactions were incubated for 15 min at 50 °C, then placed on ice and 2.5 μL of the reaction mixture was used to transform *E. coli* Stellar™ cells (Takara Bio). After overnight growth on LB-agar plate supplemented with kanamycin, colonies were screened by sequencing. The $PCP_2$-$C_3$ R2577G mutant was generated via standard Quick-Change site-directed mutagenesis procedures using primers #7 and #8 (Supplementary Table 6).

**$PCP_2$-$C_3$ SpyCatcher and $PCP_3$ SpyTag constructs**. To generate the $PCP_2$-$C_3$ SpyCatcher construct, we used an InFusion cloning reaction. The $PCP_2$-$C_3$ pOPINS plasmid was linearized using primers #21 and #22 (Supplementary Table 6), whilst the SpyCatcher insert was amplified from Addgene plasmid #35044 "pDEST14-SpyCatcher" using primers #19 and #20. The two fragments were run separately on a 0.8% agarose gel and extracted. The purified fragments were then used in an InFusion reaction according to the manufacturer's instructions (Takara Bio). Once the InFusion reaction was completed, 2.5 μL of the reaction mixture was used to transform *E. coli* Stellar™ cells (TakaraBio). After overnight growth on LB-agar plate supplemented with kanamycin, colonies were screened by sequencing.

$PCP_2$-$C_3$ SpyCatcher mutants were generated using standard Quick-Change site-directed mutagenesis procedures using primers listed in Supplementary Table 6 (#9 and #10 for H2697Q and #11 and #12 for E2702G).

To generate the $PCP_3$ SpyTag construct, the $PCP_3$ fragment was first cloned into the pHIS17 vector using an InFusion reaction. This step was necessary to introduce a His-tag at the C-terminus of the protein, thus allowing the subsequent addition of the SpyTag to the N-terminus. The pHIS17 vector was then linearized using primers #17 and #18 and the $PCP_3$ region of FscG amplified using primers #5 and #6. After the InFusion reaction was completed, 2.5 μL of the reaction mixture was used to transform *E. coli* Stellar™ cells (Takara Bio). After overnight growth

on LB-agar plate supplemented with ampicillin, colonies were screened by sequencing. A positive clone was then linearized with primers #25 and #26, while the SpyTag insert was amplified from the Addgene plasmid #35050 "pET28a-SpyTagMBP" using primers #23 and #24. After following the same InFusion and transformation procedure described above, colonies were sent for sequencing.

**Protein expression**. Production of $PCP_2-C_3$ wild-type proteins, $PCP_2-C_3$ mutant proteins (cloned in pOPIN-S vector) and $PCP_3$ proteins (cloned in pHIS17 vector) was performed as follows. A plasmid encoding the protein of interest (pOPIN-S or pHIS17) and pRARE plasmid were co-transformed into chemically competent E.coli BL21(DE3) (entdD-) cell and colonies were allowed to develop overnight at 37 °C on agar plate supplemented with the relevant antibiotics (kanamycin/chloramphenicol at a final concentration of 50 μg/mL and 34 μg/mL, respectively for the pOPIN-S/pRARE pair and ampicillin/chloramphenicol at a final concentration of 100 μg/mL and 34 μg/mL, respectively for the pHIS17/pRARE pair). Expression of all proteins was performed in 20 L TB media supplemented with the relevant antibiotic. Cells were incubated at 37 °C with shaking at 180 rpm until the $OD_{600 nm}$ reached 0.4–0.6. Protein expression was induced by the addition of IPTG (0.1 mM); cultures were subsequently grown overnight at 18 °C before being harvested by centrifugation.

**Protein purification**. All proteins in this study were purified according to the following protocol. Cells were harvested by centrifugation at $3064 \times g$ for 20 min at 4 °C. Next, the cell pellet was resuspended in Ni-NTA buffer A (50 mM Tris–HCl, pH 8.0; 300 mM NaCl; 20 mM imidazole) supplemented with protease inhibitor cocktail tablets (SIGMAFAST Protease Inhibitor Cocktail Tablets, EDTA-Free; Sigma-Aldrich) and benzonase (Sigma-Aldrich). The cells were lysed by a cell disruptor (Avestin EmulsiFlex, ATA scientific) operating at 14,000–19,000 psi, and the lysate was clarified by centrifugation at 22,680 g for 45 min at 4 °C. The supernatant was incubated at 4 °C for 1 h with 2 mL of equilibrated (Ni-NTA buffer A) Ni-NTA beads (Macherey-Nagel) with gentle stirring. After incubation, the beads were washed with 20 bed volumes of Ni-NTA buffer A. Subsequently, bound protein was eluted with 5 bed volumes of Ni-NTA buffer B (50 mM Tris–HCl, pH 8.0; 300 mM NaCl; 1 M imidazole).

For pOPIN-S derived proteins, the SUMO tag was cleaved with sentrin-specific protease (SENP) overnight while being dialyzed in a buffer composed of 50 mM Tris–HCl, pH 8.0; 300 mM NaCl, 1 mM DTT at 4 °C. The protein was subsequently incubated with 2 mL of equilibrated (Ni-NTA buffer A) Ni-NTA beads with gentle stirring for 10 min. The unbound, cleaved protein was washed with two bed volumes of Ni-NTA buffer A and used for further purification (uncut protein and the cleaved tag remain associated to the Ni-NTA beads). The protein was then incubated with 2 mL of GST agarose beads that had been previously equilibrated in PBS buffer with gentle stirring for 10 min to remove excess SENP. The unbound protein was washed with two bed volumes of PBS buffer that was then further purified.

In the case of proteins expressed with a hexa-histidine tag (pHIS17 and pET28a constructs), the tag was not cleaved. In all cases, the protein of interest was further purified after Ni-NTA purification by gel-filtration chromatography using a SRT 10 SEC 300 (105 mL) column (Sepax Technologies) connected to an ÄKTA PURE system (GE Healthcare). The column was first equilibrated with 1.2 column volumes of gel-filtration buffer (50 mM Tris–HCl, pH 7.4; 300 mM NaCl; 1 mM DTT). Subsequently, the protein was concentrated and injected onto the column, and the eluate fractionated into 1.5 mL fractions. Elution fractions containing monomeric protein were analyzed by SDS-PAGE, and appropriate fractions were combined and concentrated using centrifugal filter units (Amicon Ultra-15 centrifugal filter units (30 kDa MWCO for all $PCP_2-C_3$ constructs and 3 kDa MWCO for $PCP_3$ constructs, Merck Millipore)). Protein concentration was determined by measuring protein absorbance at 280 nm using a NanoDrop One microvolume UV-vis spectrophotometer (Thermo Scientific). Protein was concentrated to 30 mg/mL for all $PCP_2-C_3$ and 8 mg/ml for $PCP_3$ constructs, aliquoted (50 μL) into chilled 0.2 mL PCR tubes, flash frozen in liquid nitrogen, and stored at −80 °C.

**Chemical synthesis**. Unless specified otherwise, chemicals that were purchased from Sigma Aldrich, Iris Biotech, Chem-Impex International and Fisher Scientific were used without further purification. Reagent grade dichloromethane (DCM), N, N-dimethylformamide (DMF), methanol, acetonitrile (MeCN), diethyl ether, and water were purchased from Fisher Scientific.

$^1H$ NMR spectra were recorded in $D_2O$ and/or $d_4$-MeCN on the following Bruker Avance instruments: BACS-400 400 MHz or BACS600 600 MHz. NMR spectra are shown in Supplementary Figs. 21–26. High-resolution mass spectrometry (HRMS) were obtained using an Orbitrap Fusion mass spectrometer (Thermo Scientific) coupled online to a nano-LC (Ultimate 3000 RSLCnano; Thermo Scientific).

**Peptidyl-CoA synthesis**. Peptidyl-CoAs were synthesized manually on solid phase at 0.05 mmol scale with subsequent hydrazide activation and displacement to generate the desired CoA thioesters. In all, 2-chlorotrityl chloride resin (200 mg) was swelled in DCM (8 mL, 30 min), washed three times with DMF, and incubated with a 5% hydrazine solution in DMF (6 mL, 2 × 30 min). The resin was washed three times with DMF, and a solution of DMF/triethylamine (TEA)/methanol

(7:2:1; 4 mL, 15 min) was added to cap unreacted 2-chlorotrityl groups. The first Fmoc-protected amino acid (0.05 mmol) was coupled to the resin overnight using O-(6-chlorobenzotriazol-1-yl)-N,N,N′,N′-tetramethyluronium hexafluorophosphate (HCTU, 0.05 mmol) and diisopropylethylamine (DIPEA, 0.05 mmol). After that, unreacted hydrazine groups were capped with Boc−glycine (0.15 mmol) that had been activated prior to addition using HCTU (0.15 mmol) and DIPEA (0.15 mmol) for 1 h. Subsequent Fmoc removal was performed using a 20% piperidine solution in DMF (3 mL, 3 × 30 s) followed by coupling of the desired Fmoc- or Boc-protected amino acid (0.15 mmol) after pre-activation with HCTU (0.15 mmol) and DIPEA (0.15 mmol) for 1 h. Cleavage of the hydrazide peptide from resin and removal of side chain protecting groups was accomplished using trifluoroacetic acid/triisopropylsilane/water (TFA/TIS/$H_2O$, 95:2.5:2.5 v/v′/v″, 5 mL) with shaking at room temperature for 1.5 h. The resin was removed by filtration and washed twice with TFA. The filtrate was then concentrated under a stream of $N_2$ to ~1 mL, the peptide precipitated with ice-cold diethyl ether (~9 mL) and collected by centrifugation in a flame-resistant centrifuge. The crude peptide was purified using preparative RP-HPLC (using a gradient of 0−40% MeCN over 30 min). Purified hydrazide peptides were then dissolved in buffer 1 (6 M urea and 0.2 M $NaH_2PO_4$, pH 3) to a final concentration of 5 mM. The solution was cooled to −15 °C using a salt/ice bath, 0.5 M $NaNO_2$ (0.95 eq.) was added and the mixture was stirred for 10 min. CoA (1.2 eq., dissolved in buffer 1) was then added to the reaction. The pH was slowly adjusted to 6.5 using $KH_2PO_4/K_2HPO_4$ buffer (6:94 v/v 1 M, pH 8.0). The reaction mixture was stirred at −15 °C for additional 2 h, before the final peptidyl-CoA product was purified using preparative RP-HPLC (gradient 0−40% MeCN over 30 min)[39,40]. For characterization see Supplementary Figs. 18−19 and 24−25.

**Stabilized aminoacyl-CoA synthesis**. CoA (1 eq.) was dissolved in 10 mL of buffer 2 (0.02 M ammonium bicarbonate and 6.5 mM EDTA, pH 8). Tris (2-carboxyethyl)phosphine (TCEP, 1.2 eq.) was added and the mixture stirred for 30 min. Alkyl bromide (3 eq.) was dissolved in MeCN (2 mL) and added to the CoA solution, which was then stirred at room temperature overnight. The desired compound was concentrated and purified by preparative RP-HPLC purification (MeCN gradient 0–40% over 30 min)[41]. For characterization see Supplementary Figs. 20–21 and 26–27.

**Aminoacyl-CoA synthesis**. Boc-amino acid (2 eq.), TEA (2 eq.) and (1-Cyano-2-ethoxy-2-oxoethylidenaminooxy)dimethylamino-morpholino-carbenium hexa-fluorophosphate (COMU, 2 eq.) were dissolved in DMF and stirred in an ice bath for 30 min before the dropwise addition of a solution of DMF containing CoA (1 eq.). The mixture was then stirred overnight at room temperature. Crude Boc-aminoacyl-CoA was precipitated by the addition of ice-cold $Et_2O$ and the pellet collected using centrifugation in a flame-resistant centrifuge. The addition of $Et_2O$ and subsequent centrifugation was repeated three times to wash the sample. The crude product was purified by preparative RP-HPLC (MeCN gradient 0-40% over 30 min). Cleavage of the Boc group was performed using a mixture of TFA/ TIS/ $H_2O$ (95:2.5:2.5, v/v′/v″; 1 mL) for 1 h and the solution was concentrated under a stream of $N_2$ before precipitation of the peptide was performed by addition of ice-cold $Et_2O$, followed by subsequent washing (3x)[33]. For characterization see Supplementary Figs. 22–23 and 28–29.

**Preparative HPLC**. Compound purification was performed using a Shimadzu High Performance Liquid Chromatograph equipped with a SPD-M20A Prominence Photo Diode Array Detector and two LC-20AP pumps. Purification used a Waters XBridge BEH300 Prep C18 column (5 μm, 19 × 150 mm) at a flow rate of 10 mL/min. The solvents used were water + 0.1% TFA (solvent A) and ACN + 0.1% TFA (solvent B).

**PCP-domain loading**. All proteins containing PCP-domains were expressed and purified in their apo form, which were converted into their holo form using the phosphopantetheinyl transferase Sfp (R4-4 mutant) and desired CoAs[32]. The loading reaction utilized a 1:2:0.1 molar ratio of the PCP domain, peptidyl-/aminoacyl-CoA and Sfp (R4-4 mutant), respectively. Peptidyl-CoA (200 μM) was loaded onto the PCP-containing construct (100 μM) for 1 h at 30 °C using the Sfp (10 μM) in PCP-loading buffer (50 mM HEPES, pH 7.0; 50 mM NaCl; 10 mM $MgCl_2$). After the loading reaction, the remaining peptidyl-CoA was removed by three concentration/ dilution steps using centrifugal concentrators (Amicon® Ultra-0.5 mL centrifugal filters units (30 kDa MWCO for $PCP_2-C_3$ Constructs (also removing Sfp) or 3 kDa MWCO for $PCP_3$ constructs, Merck Millipore) in gel-filtration buffer (50 mM HEPES, pH 7.4; 300 mM NaCl, 1 mM DTT). Holo-PCP constructs were then immediately used for in vitro reconstitution assays or crystallization experiments.

**In vitro reconstitution of NRPS**. The peptide loaded $PCP_2-C_3$ Spy-Catcher construct was incubated with unloaded $PCP_3$ Spy-tag construct (both 100 μM) for 10 min at 30 °C, which was followed by loading of the desired aminoacyl-CoA on the $PCP_3$ as described above. The reaction was then incubated for an additional 1 h at 30 °C to allow for the condensation reaction to occur. For thioether tethered amino acid loaded $PCP_3$ substrates, reaction mixtures were directly analyzed using

nano LC ESI MS (see below Ppant ejection section)[42]. For thioester-tethered amino acid loaded $PCP_3$ substrates, chemical cleavage by an addition of 15 μL of methylamine liberated the methylamide peptides; reaction mixtures were incubated for 15 min at room temperature. The peptide products were then purified from the reaction mixture using solid phase extraction (Strata™-X-33 μm Polymeric Reversed Phase Tubes; 30 mg/mL; Phenomenex). Before loading the sample, cartridges were activated with 0.1% formic acid (FA) in methanol (1 mL) and subsequently equilibrated with 0.1% FA in water. Samples were loaded onto equilibrated cartridges and the solution passed through the column bed under gravity. Once the samples were loaded, the cartridge was washed with 0.1% FA in water (1 mL) before the peptides were eluted with 0.1% FA in MeCN/water (50/50, v/v). The samples were then dried by freeze dryer at −50 °C and analyzed by HRMS.

**HRMS and $MS^2$ measurements**. High-resolution mass spectrometry measurements were performed on an Orbitrap Fusion mass spectrometer (Thermo Scientific) coupled online to a nano-LC (Ultimate 3000 RSLCnano; Thermo Scientific) via a nanospray source. Peptides were separated on a 50-cm reverse-phase column (Acclaim PepMap RSLC, 75 μm × 50 cm, nanoViper, C18, 2 μm, 100 Å; Thermo Scientific) after binding to a trap column (Acclaim PepMap 100, 100 μm × 2 cm, nanoViper, C18, 5 μm, 100 Å; Thermo Scientific). Elution was performed on-line with a gradient from 6% MeCN to 30% MeCN in 0.1% formic acid over 30 min at 250 nL min⁻¹. Full scan MS was performed in the Orbitrap at 60,000 nominal resolution, with targeted $MS^2$ scans of peptides of interest acquired at 15,000 nominal resolution in the Orbitrap using HCD with stepped collision energy (24 ± 5% NCE). QualBrowser (XCalibur 3.0.63, Thermo Scientific) was used to view spectra and generate extracted ion chromatograms for the singly charged species at 20 ppm. The level of peptide extension in the assays shown in Fig. 6 were calculated using the following formula: percentage conversion = peak area (product)/(peak area (donor) + peak area (product)) × 100. Predicted $MS^2$ fragments were generated with MS-Product (ProteinProspector v5.22.1, UCSF) and manually assigned to spectra[43]. See Supplementary Figs. 30–35.

**Ppant ejection**. Mass spectrometry measurements were performed on a Micro-TOFq mass spectrometer (Bruker Daltonics) coupled online to a 1200 series capillary/nano-LC (Agilent Technologies) via a Bruker nano ESI sprayer. Proteins were separated on a 150-mm reverse-phase column (ZORBAX 300SB-C18, 3.5 μm, 0.075 × 150 mm; Agilent Technologies) after binding to a trap column (ZORBAX 300SB-C18, 5 μm, 0.30 × 5 mm cartridges; Agilent Technologies). Elution was performed on-line with a gradient from 4% MeCN to 60% MeCN in 0.1% FA over 30 min at 300 nL/min. Proteins >20 kDa were separated on a MabPac SEC-1 5 μm 300 Å 50 × 4 mm (Thermo Scientific) column with an isocratic gradient of 50% MeCN, 0.05% TFA and 0.05% FA at a flow rate of 50 μL/min. The protein was eluted over a 20-min run-time monitored by UV detection at 254 nm. After 20 min the flow path was switched to infuse Low concentration Tune mix (Agilent Technologies, Santa Clara, CA, USA) to calibrate the spectrum post acquisition. The eluent was nebulized and ionized using the Bruker electrospray source with a capillary voltage of 4500 V dry gas at 180 °C, flow rate of 4 L/min and nebulizer gas pressure at 0.6 bar. MSMS spectra were acquired by manual selection of isolation mass and isolation width with a collision energy of 32. The spectra were extracted and deconvoluted using Data explorer software version 3.4 build 192 (Bruker Daltonics, Bremen, Germany). For analysis see Supplementary Fig. 13.

The HRMS and PPant ejection data have been deposited to the ProteomeXchange Consortium via the PRIDE[44] partner repository with the dataset identifier PXD024004.

**Crystallization of $PCP_2$-$C_3$ proteins**. Aminoacyl-CoAs were loaded onto $PCP_2$-$C_3$ affording the *holo* forms of $PCP_2$-$C_3$ and concentrated to a final concentration of 30 mg/mL in gel-filtration buffer. Initial screening was performed at the Monash Molecular Crystallisation Facility (MMCF) with subsequent optimization performed in 48-well sitting-drop plates. Crystallization trials of $PCP_2$-$C_3$ at a concentration of 30 mg/mL in a 1:1 ratio (v/v) with the crystallization solution (2 μL drops) led to a condition composed of 18–22% v/v PEG 3350 and 0.17–0.3 M magnesium); crystals formed overnight at room temperature. Crystals were cryoprotected by transferring in a drop made of the reservoir solution supplemented with glycerol (to a final concentration of 30% v/v). Crystals were collected in cryoloops and flash frozen in liquid nitrogen.

**Crystallization of $PCP_3$ protein**. Initial screening was performed at the Monash Molecular Crystallisation Facility (MMCF) with subsequent optimization performed in 48-well sitting-drop plates (MRC Maxi plate (molecular dimensions)). After optimization, the best crystallization condition was composed of 500 μM Bis-Tris, pH 5.5, 1.8 M $NH_3SO_4$. Sitting drops were made of 1 μL of $PCP_3$ at a concentration of 11 mg/mL and 1 μL of the crystallization solution. Crystals formed overnight at room temperature. Crystals were cryoprotected by transferring in a drop comprising reservoir solution supplemented with glycerol (to a final concentration of 30% v/v) and flash frozen in liquid nitrogen.

**Data collection and structure determination**. All datasets were collected at the Australian Synchrotron (Clayton, Victoria, Australia) on beamlines MX1[45] (R2577G $PCP_2$-$C_3$ PPant, WT $PCP_2$-$C_3$ Gly$_{stab}$ and $PCP_3$: wavelength 0.95372 Å) and MX2 (WT $PCP_2$-$C_3$ PPant; wavelength 0.95374 Å) equipped with an Eiger detector (Dectris) at 100 K[46]. Data processing was performed using XDS[47] and AIMLESS as implemented in CCP4[48]. Phases for the $PCP_2$-$C_3$ constructs were obtained from a single wavelength anomalous diffraction experiment (SAD) using xenon-derivatized crystals. In brief, crystals were mounted into a cryo-loop and briefly exposed to xenon gas using the Hampton Research Xenon Chamber available at the Australian Synchrotron and flash frozen in liquid nitrogen. The SAD dataset was then reduced with XDS[47] and the phases obtained using HKL2MAP[49]. The initial model generated by HKL2MAP was subsequently used in molecular replacement experiments to obtain phases for the other datasets using PHENIX in-built Phaser module[50]. The crystals belonged to the $P2_12_12_1$ space group, with the unit cell comprising 2 highly similar copies of the $PCP_2$-$C_3$ construct (Supplementary Table 1). His-$PCP_3$ crystals belonged to the $P4_3 3 2$ space group, with one single subunit per cell. Phases were obtained in a molecular replacement experiment using a model generated by iTasser[51] and performed within the in-built Phaser module in PHENIX.

Structural models were built and refined using COOT[52] for model building and PHENIX-refine for refinement[50]. Ramachandran statistics (favored/disallowed): WT $PCP_2$-$C_3$ PPant (97.8%, 0%), R2577G $PCP_2$-$C_3$ PPant (97.1%, 0.1%), WT $PCP_2$-$C_3$ Gly$_{stab}$ (97.7%, 0.1%), $PCP_3$ (100%, 0%). The model quality of each structure assessed by Molprobity (score/percentile): WT $PCP_2$-$C_3$ PPant (1.03, 100th), R2577G $PCP_2$-$C_3$ PPant (1.27, 99th), and WT $PCP_2$-$C_3$ Gly$_{stab}$ (1.00, 100th) $PCP_3$ (1.24, 100th). Similar structures were identified by DALI[53], and the PCP/C-domain interface analyzed using PISA[54]. All graphics were generated with Pymol (Schrödinger LLC) or UCSD Chimera[55].

**Database search**. All sequences used for statistics and correlation analyses were isolated from the MiBiG database[56], accessed on 03.03.2020. Domain sequences, specificities and other information were extracted from the MiBiG entries' gbk and json files by parsing for keywords. We identified 2049 C-domains with known selectivity (1456 $^LC_L$ and 593 $^DC_L$), of which downstream A domain specificity was known in 488 sequences. The C-A linkers, which include the possible "pocket" residues were defined as the sequences that start at the end of a C domain and end at the beginning of the downstream A domain within the same NRPS gene. 401 such regions with known A domain specificity were isolated and used in the corresponding analysis. The conserved HHxxxDG motifs were analyzed from 481 C domain sequences with known downstream A domain specificity, this time including starter C domains and ones with dual selectivity.

**Correlation and statistical analyses**. All Multiple Sequence Alignments (MSAs) used for statistics and correlation analyses were produced with the MUSCLE[57] tool (version 3.8.31) with default settings. Sequence logos (Supplementary Figs. 6 and 11) were created with WebLogo (version 2.8.2 or 3.7)[58]. We studied the correlation between the "pocket" residues of the $PCP_2$-$C_3$ construct and the acceptor substrate with two methods. The correlation coefficient Spearman's rho was computed with the spearmanr function (scipy.stats module) of the SciPy[59] python library (version 1.4.1), by using the sum of molecular weights of the "pocket" residues and the mass of the acceptor substrate. Principal Component Analysis (PCA) was conducted with the PCA function (sklearn.decomposition module) of the scikit-learn[60] python library (version 0.22.2.post1) while taking into account each residue's molecular mass. The PCA graph (Supplementary Fig. 15), as well as the stacked barplots (Supplementary Fig. 16) were visualized with functions from the matplotlib.pyplot module of the Matplotlib[61] python library (version 3.2.1) and from the NumPy[62] python library (version 1.18.1). All scripts were implemented with Python version 3.7.6.

**CAVER analysis**. Protein tunnels were identified and assessed using Caver 3.0[63] using a probe radius of 0.7 Å, shell radius of 4 Å, and shell depth of 4 Å. The clustering threshold was set at 3.5. The starting point was defined as the point between His2697, Glu2702, and Pro2841.

**Computational protein–protein docking, substrate docking, and molecular dynamics simulations**. Computational protein–protein docking, substrate docking, and molecular dynamics (MD) simulations were performed using Schrödinger Release 2019-1. In all cases, protein structures were prepared using the Protein Preparation Wizard in Maestro. Following pre-processing of the pdb files (including addition of hydrogens), all water molecules and small molecules were removed, and alternate confirmations were restricted to the most probable rotamer. Hydrogen bonds were optimized using ProPKA3[64,65] at pH 7.0, and the restrained minimization was performed using the OPLS3e force field[66] (converging heavy atoms to RMSD of 0.30 Å).

**Computational protein–protein docking of the $PCP_3$-domain**. Protein–protein docking between the $PCP_3$ and the $C_3$ domain from the unloaded $PCP_2$-$C_3$ didomain structure (chain A, residues 2558–2999) were performed using the protein–protein docking wizard in Maestro, which uses the PIPER docking algorithm[67]. In order to constrain the docking of $PCP_3$ to the acceptor PCP binding site, a distance restraint was set between Ser3558 of the $PCP_3$ and Tyr2585 of the $C_3$ domain (minimum 2 Å,

maximum 15 Å). During rigid-body protein-protein docking, 70000 ligand rotations were probed, and the top 30 poses were refined prior to analysis.

**Molecular dynamics simulations of C₃ domain**. Molecular dynamics (MD) simulations were performed in Desmond (Schrödinger Release 2019-1). Simulations were initiated from the structures of the $C_3$ domain (chain A, residues 2558–2999) from unloaded $PCP_2$-$C_3$ didomain and $Gly_{stab}$-$PCP_2$-$C_3$ didomain structures. Protein structures were prepared using the Protein Preparation Wizard as described above, then placed in an orthorhombic box with a buffer of 10 Å around the protein molecule and periodic boundary conditions (PBCs) were applied. This provided sufficient distance between neighboring protein molecules once PBCs were applied (~20 Å); this distance was significantly larger than the 9 Å electrostatic cut-off used during simulations. Each system was solvated with SPC water molecules, and the system was neutralized through the addition of $Na^+$ ions. Following the default Desmond relaxation protocol, 100 ns production runs were performed in triplicate using the default Desmond settings. Snapshots were recorded every 0.5 ns and were analyzed using CAVER 3.0, as described above. Dihedral angles of Arg2577 were measured using the simulation event analysis wizard in Schrödinger. The OPLS3e force field[66] was used at all stages of the simulation. The OPLS3e force field is the default force field in Desmond and performs well against other force fields for the simulation of protein molecules[66].

**Computational docking of substrates into C₃**. Docking of PPant-linked substrates was performed in Schrödinger using the ligand docking wizard and Glide algorithms[68]. The $C_3$ domain from the $Gly_{stab}$-bound $PCP_2$-$C_3$ didomain structure (chain A, residues 2558 – 2999) was used as the receptor for docking studies and prepared as described above. The PPant from the $Gly_{stab}$-bound $PCP_2$-$C_3$ didomain structure was used as a template from which alternate substrates were modeled. Alternate ligands were constructed using the 3D builder tools in Maestro. The LigPrep wizard was used to prepare these ligands using the OPLS3e force field and possible ionization states (pH 7.0 ± 2) were generated using Epik. Ligands were computationally docked using the "standard protocol" option in the ligand docking tool. The central phosphorous of the phosphate moiety was restrained to the position of the phosphorus in the structure of the $PCP_2$-$C_3$ didomain in complex with $Gly_{stab}$ (sphere of radius 3 Å around this position).

The E2702G mutant was modeled in silico; using the $C_3$ domain from the $Gly_{stab}$-bound $PCP_2$-$C_3$ didomain structure (chain A, residues 2558 – 2999) as a template, the mutation was introduced using the mutation tool in Maestro. A basic local minimization step was performed, followed by Protein Preparation Wizard's restrained minimization, as described above.

**Density functional theory**. Density functional theory (DFT) computations were performed in Gaussian 16[69]. The B3LYP-D3 functional[70–74] and 6-31 G(d) basis set were used, in conjunction with the SMD model[75] of implicit diethyl ether ($\varepsilon = 4.24$, chosen to approximate the dielectric constant of the interior of an enzyme). Transition states were characterized by the presence of a single imaginary vibrational frequency corresponding to the reaction coordinate. Intrinsic reaction coordinate[76,77] calculations were also performed to identify the local minima situated on either side (reactant and product) of transition states. The DFT computations were carried out to determine whether the attack of the amine on the thioester has an intrinsic preference for a stepwise or concerted mechanism. They did not attempt to model the exact binding orientation of the substrates within the enzyme active site.

**Reporting summary**. Further information on research design is available in the Nature Research Reporting Summary linked to this article.

## Data availability
Crystal structures have been deposited to the protein databank (PDB) under the accession numbers 7KVW, 7KW0, 7KW2, and 7KW3. HRMS and PPant ejection data have been deposited to the ProteomeXchange Consortium via the PRIDE partner repository with the dataset identifier PXD024004. All sequences used for statistics and correlation analyses were isolated from the MiBiG database [https://mibig.secondarymetabolites.org/]. Source data for Fig. 5d, Supplementary Fig. 5c, and Supplementary Fig. 5f are provided with this paper. Source data are provided with this paper.

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

## Acknowledgements

J.Yin (University of Chicago) for the R4-4 Sfp expression plasmid; M. Kaniusaite (Monash) for assistance with cloning; T. Harshegyi, L. Scully, and S. Stamatis (Monash) for assistance with protein purification; D. Maksel and G. Kong (MMCF, Monash) for assistance with crystal screening experiments. This research was undertaken on the MX1 and MX2 beamlines at the Australian Synchrotron, part of ANSTO, and made use of the Australian Cancer Research Foundation (ACRF) detector. We would like to thank the beamline scientists at the Australian Synchrotron for their support during data collection. Computational resources were provided by the National Facility of the Australian National Computational Infrastructure through the National Computational Merit Allocation Scheme and by the University of Queensland Research Computing Centre. This work was supported by Monash University, EMBL Australia, the Australian Research Council (Discovery Project DP180103047, DP190101272, and DP210101752) and the National Health and Medical Research Council (APP1140619 to M.J.C.). T.I. is grateful for the support of the CASS foundation (grant #8583). A.G. is grateful for the support of the Deutsche Forschungsgemeinschaft (DFG; Project ID # 398967434-TRR 261). This research was conducted by the Australian Research Council Centre of Excellence for Innovations in Peptide and Protein Science (CE200100012) and funded by the Australian Government.

## Author contributions

The study was designed by T.I. and M.J.C. All cloning and protein purification was performed by T.I. and Y.T.C.H. Structural analysis was performed by T.I., Y.T.C.H., and M.J.C. with insightful contributions from G.L.C. and J.A.K. Chemical synthesis was performed by Y.T.C.H. and condensation assay was performed by Y.T.C.H. and T.I. Turnovers results were analyzed by Y.T.C.H. and D.L.S., with M.T. assisting with analysis of HRMS experiments. HRMS and protein MS measurements were performed by D.L.S., R.J.A.G., and R.B.S. Computational docking and molecular dynamics simulations were performed and analyzed by J.A.K. and C.J.J. Bioinformatics and correlation analyses were performed by A.G. and N.Z. Computational analysis was performed by K.H.C. and E.H.K. The manuscript was written by T.I., Y.T.C.H., and M.J.C. with input from the other authors.

## Competing interests

G.L.C. is a co-director of Erebagen Ltd. All the other authors declare no competing interests.
