## [Peer Review File · Nature Communications]

REVIEWER COMMENTS

Reviewer #1 (Remarks to the Author):

The manuscript by Izore and colleagues describes the crystal structure of the condensation domain from the fuscachelin A biosynthetic protein FscG. The authors describe several structures of the PCP2-C3 didomain construct as well as the PCP3 domain. Importantly, the asymmetric unit of the didomain structures contains two independent molecules in which the PCP of one chain is docked in the acceptor site of the condensation domain of the other molecule. Thus, the structure illustrates the positioning of the holo-PCP in the acceptor site of the condensation domain. While other similar structures have been determined, the critical advance of this manuscript is that the authors have loaded the PCP with the amino acyl thioether analog of the pantetheine, thus demonstrating for the first time the structure of a holo PCP bound in the condensation domain acceptor site that is loaded with its amino acid substrate.

This allows the authors to make several observations. First, the authors use bioinformatic analysis to demonstrate that there is not an obvious "specificity pocket", in contrast to what is seen with the NRPS adenylation domains. Additionally, the authors explore the role of several residues in the pantetheine pocket and the active site. Arg2577 is proposed to play a role in gating the pantetheine tunnel to control access to the active site. Glu2702, which replaces the canonical glycine of the C-domain HHxxxDG motif, is proposed to position the acceptor amino acid in the condensation domain for small amino acid substrates. The more common glycine is used with larger amino acids, where interactions with the substrate side chains can probably better position the α -amino group for the condensation reaction.

This paper is an important addition to our knowledge of the structural mechanisms of NRPS condensation domains. The structures are well determined and supported by functional and bioinformatic analysis. I have a few suggestions the authors may wish to consider to help clarify the writing and analysis in a few places.

Pg 4. The sentence within the last paragraph of the introduction "The C domain is shown to be tolerant of a range of aliphatic acceptor amino acid acceptor substrates, with the limited acceptance rationalized..." is a bit of a contradiction (tolerant/limited acceptance). Perhaps include the word "small" to read "...tolerant of a small range ...". (Also note duplication of "acceptor")

Figure 2 and Pg 7, line 1. I tend to think of symmetry-related in crystals to refer to interactions between asymmetric units. Thus I do not think that molecule B should be highlighted as "symmetry" nor described as docked to a "symmetry-related" C3 domain (pg 7). The description of Pg 5, 3 lines from the bottom, is more accurate.

Figure 4. Apo and holo are generally used to describe a protein +/- a cofactor. Thus, using "apo" in panels A and B is not accurate, with "unloaded" being correct (as in the legend). Please also list contour level and any carve radius in the legend.

Figure 5, panels A and B. Glu2702 is mislabeled as 2707. This is incorrect in the legend as well.

Pg 11. The authors should provide a few sentences to provide a brief description of the Spycatcher/Spytag system so that the reader does not need to go the reference to know whether this a short peptide, two large protein domains, or a biotin/streptavidin based system.

Pg 13. " zwitterioinic"

Supplementary Material

A table of contents for the SI can help guide the reader.

Table S1. The Resolution entry for all four datasets is formatted as "Low Resolution limit (High Resolution limit)". While everyone would understand what is meant, the format is a bit unusual as in every other entry of the table, the entry not in parentheses is the data for the complete data set and the entry in parentheses is the highest resolution shell. Resolution data should simply be presented as "48.14 – 2.18" or if preferred "48.14 – 2.18 (2.3 – 2.18)".

Table S4. Please define $\Delta \rho$

Figure S1. Please state in the legend whether the aligned halves are the N-terminal or C-terminal subdomains.

Figures S8, S9, and S10, detailing the plausible mechanisms from the supplementary discussion are not present in the SI but rather are part of Figure 5 of the main text.

Please consider expanding the Legend to Figure S14, including a label for the X-axis. I understand the motivation of this figure and believe I understand how the information is presented but it took me a while to interpret this.

Andrew M. Gulick
University at Buffalo

Reviewer #2 (Remarks to the Author):

This is a paper that reports on the structure of an NRPS module with the aminoacyl group (stable analog) bound. This is one of the missing structures in the understanding of NRPS assemblies. It was also so good to see that PPant ejection still being used. I discovered it in 2007 and published it in 2008 before starting my own career so it brings back some wonderful memories.

It was an enjoyable read. Some comments and questions. All are minor and even if the authors have no answer, the paper would still be appropriate for NComms.

Fig 1a the lighter colors and letters are hard to see for the structures in the NRPS assembly lines. I suggest making a) the width of the entire page. And move b) and c) around to accommodate.

Figure four it would be nice to see an overlay of the apo vs acylated PPant to show how the configurations stayed the same and where they are dissimilar.

This is a bit of an open-ended question but this could have a big impact if there was such a trend and it may be possible to answer these questions with this unique structure. Are there conserved residues in the C domain that bind PPant?, are there conserved residues in the C domain that define Acyl selectivity? I see that you did a pocket analysis. It is not entirely clear what this is and what the training set was but perhaps look for two details. 1) are there specific residues associated with the active site depending on the substrate that is loaded. 2) are there polarities of residues that define substrate specificities. The tricky part is that you have to half the active site in the donor and acceptor region and then ask these questions. It may be that this is currently not possible to identify if there is a true donor or acceptor region in the C domain with the currently available structures but if it is then it could be very interesting.

"This forced us to explore alternatives to thioester-tethered amino acids, and we chose to use an analog of the aminoacyl-CoA with a thioether in place of the reactive thioester." I see from the structure that the carbonyl is also missing, can the authors clarify what the actual non-hydrolyzable thioether is in this section? (I saw the structures are shown in the SI but would be nice to mention what the mimic is in this section.) Perhaps just change to the following "...a thioether, missing the reactive carbonyl, that makes the thioester susceptible to nucleophiles."

Regarding HHXXDG vs HHXXDE, is there a trend among substrates that are connected based on sequence gazing the understanding of the new structure and the type of amino acids or acids that are coupled? I know you tested the E to G mutation in vitro but in vivo do you think the condensations become more promiscuous? I.e. is this another gatekeeper for selectivity for condensation build into the system post A domain activation? I know the discussion has a description of selectivity but limited in the context of the condensation rxn.

Figure S11, I would not call it HRMS. As the resolution is quite poor compared to FT instruments and the isotopes for proteins are not detected. The mass accuracy is OK not amazing but works. Maybe just specify the resolution of the MS instead so you don't have to specify HRMS. The numbers of the masses have one too many sig figs for the PPant ejections. For example, 621.3177, found: 621.3049 this is 0.128 Da difference. This should be 621.32, found: 621.31 based on this mass accuracy. The deuterated experiment is a nice confirmation that eliminates the need for the super high mass accuracy.

The paper does not have a data availability section in the paper. Coordinates, MS data, etc.

Of course, due to my personal bias I would also encourage to make the MS/MS spectra of all the synthetic compounds publicly available not just as copies in a PDF.

Congrats, it was a joy to read and stay healthy, Pieter Dorrestein.

Reviewer #3 (Remarks to the Author):

The paper by Izoré et al. presents a state-of-the-art multidisciplinary research of C-domain structure and selectivity in non-ribosomal peptide biosynthesis. Four novel crystallographic structures are reported with a resolution of ~2 Å. Expert analysis of these structures followed by biochemical characterization, site-directed mutagenesis, bioinformatic analysis and molecular modeling provided novel insights into the roles of key residues of the C-domain. Most notable is the new understanding of the role of the HHxxxDE pattern and R2577. A crystallographic structure of C-domain in complex with an acceptor PCP-domain bearing a substrate is reported for the first time, thus shaping further steps towards our understanding of structure-function relationship in NRPSs and related proteins.

The take-home message of the presented manuscript is clear. I believe, the new crystallographic structures enriched by extensive multidisciplinary analysis have the capacity to make an immediate contribution to mechanistic understanding of non-ribosomal peptide biosynthesis, though some parts of the work require clarification or revision.

My criticism is focused on the handling of the aspect of conformational plasticity of NRPSs and their complexes. As authors clearly state in the introduction "NRPSs are highly flexible and the interactions between individual domains change during the process of chain assembly". In this context, my major points are as follows:

1. First and foremost, if NRPSs and their complexes are known or expected to be flexible, then how was it even possible to characterize the reported PDB structures? What novel approach was used in

the work that made it possible to characterize such complexes?

2. If NRPSs and their complexes are flexible, then I would expect a series of conformational variants to exist in equilibrium. Then, authors also continuously refer to their PDB models as "structural snapshots". What conformational and functional states of the complexes are reflected in the reported PDB "snapshots" compared to the conformation ensemble of NRPSs and their complexes?

Bioinformatic comparison of the obtained complexes with related PDB structures and also molecular modeling was reported, mostly in Supplementary material, but I did not immediately find the answer to my questions. Thus I believe this issue was not address sufficiently clear and should be presented, possibly, as a dedicated section. How flexible are the reported complexes? How representative these "static" models are in "dynamic" context of conformation ensemble of NRPSs and their complexes?

Would they move/rearrange during MD simulation? The molecular modeling part left the impression that the PDB complexes were stable during MD. How does this conclusion coexists with the notion of NRPSs being highly flexible?

3. Authors report molecular docking and transition state calculations. These are well-known and widely used approaches, but they all demand a single "static" high-quality model as input. In practice, this usually means that the protein in question should be relatively "rigid". I had an impression that authors used the original "static" PDB models as input to these methods. Since NRPSs and their complexes are considered highly flexible, I wonder about the validity of such approach. Either there is a justification for use of "static" models of the complexes in question for docking and QM, or additional work should be performed, e.g. docking into several structural states and snapshots obtained from PDB or molecular dynamics.

Other issues:

1. Regarding the PDB structures

The four crystallographic models present primary result of the work, and yet the respective PDB codes were not mentioned even once in the main text. Instead, the codes and explanation of their contents are buried deep in the Supplementary data. I would like to see the PDB codes and a brief explanation of contents of the respective 3D-structures in a dedicated section of the main text. The figures obtained using the original models should also cite their codes. I even suggest to additionally provide a list of the codes in the abstract, for convenience of the users.

Then, unless there is some specific rule against it that I am not aware of, I would like to see all four PDB structures (i.e. files which I can load into PyMol and view in 3D) before submitting my final decision. In recent years, there is a growing amount of PDB entries containing obvious human-visible errors. The formal validation reports are interesting, but without the PDB models themselves they do not show the full picture. So I believe an extra check of PDB files by independent pair of eyes will be useful to both the authors and further users of this data.

2. Bioinformatic analysis

A multiple sequence alignment tool was used to superimpose sequences of individual domains for bioinformatic analysis. Sequence alignment is limited to relatively close homologs with a high level of pairwise sequence similarity (at least ~30-40% is needed to hope for a meaningful comparison, as everything below 40% is commonly considered as the "twilight zone" of sequence alignment). However, the level of sequence identity between the aligned sequences was not stated in Supplementary material (Bioinformatic methods section). The level of pairwise sequence similarity should be stated. If it resides below 40%, other methods should probably be used (e.g. 3D-alignment, profile-based alignment, structure-guided sequence alignment, etc.), or additional justification should be provided.

Authors discuss sequence conservation statistics in the manuscript (e.g. 80% of Gly is found in DCL domains in positions equivalent to R2577, etc.). It usually makes sense to calculate such statistics after the removal of redundant, duplicated, incomplete, etc. sequences (i.e. the redundancy filter). No such filter was stated in methods, as it seems 1456 LCL and 593 DCL sequences were collected as is, and may contain redundant information. The same concern also applies to the calculation of correlation coefficients.

3. Molecular modeling

I am concerned by the Molecular dynamics protocol. An orthorhombic box of 10 Å around the protein molecule was used for simulation of a potentially flexible protein-protein complex. Such a small gap between the edge and the box means that anticipated relative movement of stacked subunits during the simulation would cause them to interact with their images in neighboring cells (assuming that periodic boundary conditions were used, which, I think, was not clearly stated, but implied by the protocol). One of many potential side-effects of such small-box setup could be artificial over-stabilization of the complex. If indeed the authors expect their complexes to implement intrinsic conformational flexibility, a box with 30Å or so should probably be used instead. Even such a huge box would not be sufficient if subunits move too much, but in the absence of significant major rearrangements in the protein-protein complexes such box would do to present a valid simulation, ruling out artificial over-stabilization. Authors should clarify their simulation protocol and the choice of this crucial parameter, or perform additional simulations.

I wonder what was the rationale for selecting OPLS3e force field for modeling, given the two key aspects of the study: (1) the need to model the expected conformational plasticity and (2) the need to model crucial interactions involving charged residues (R2577 in particular)?

ProPKA version should be stated, as different releases of this particular software are known to give significantly different results.

The "Results" section of the main text mentions the term "docked" / "docking" multiple times. The authors should clarify the meaning of the term, as it was not immediately clear to me. Does it refer to a predictive computational method called "molecular docking", or does it refer to a "stacking" of protein subunits observed in crystallographic complexes obtained experimentally.

There is a section called "Supplementary Discussion" discussing a computational investigation of the catalytic mechanism. I was neither able to quickly find a reference to this section in the main text, nor the QM calculations were discussed in the Methods. Thus, I was not able to immediately understand the value of this supplementary section, its results and relevance to the study.

Reviewer #4 (Remarks to the Author):

In "Understanding condensation domain selectivity in non-ribosomal peptide biosynthesis: structural characterization of the acceptor bound state", Cryle and colleagues present a nice series of structures of a PCP-C didomain, in which the PCP domain is docked at the acceptor site of the adjacent C domain. Structures in which the PCP is loaded with apo PPant show this moiety to curl away from the active site, but when a nearby R2577G is mutated, or when a propylamine representing glycine is attached, the PPant enters the active site.

The authors set up a useful PCP2-C3-PCP3 system with Spycatcher to specifically load PCP2 and PCP3 with different PPant moieties and use it to present Gly, Ala, Leu or Phe to the C domain, and to analyze three C domain mutations.

Overall, the manuscript is well written and scientifically sound. It is not clear from the data presented whether some specific points, such as the putative gating role for R2577, are a general feature of NRPS biology, but this paper will be a nice addition to the NRPS literature.

Specific comments and questions:

About R2577:

SI Figure S6: R2577 is conserved in 70% of LCL domains. Can the authors comment on why they think the next most common residues are G, A, Q and S? Also, what is the conservation in other C-type domains, for example Cyc domains or starter C domains? It seems odd that this is not conserved in DCL domains given that the acceptor substrate has L chirality.

P8: "LCL*" What does the asterisk signify here?

P10: "R2577 now forms specific interactions with two of the carbonyl oxygen atoms in the Ppant arm (3.7 Å and 3.8 Å)"

These are long distances and would be quite weak interactions.

P13: "One hypothesis for the role of this residue would be to prevent the unwanted "pass-through" of donor substrates without elongation (e.g. from PCP2 to PCP3)."

Is pass-through a likely event, given that the nucleophile in the pass-through reaction (PPE thiol) is 3 atoms away from the nucleophile peptide bond formation (amino group).

If this were an important mechanism, pass through should be observable in the R2577G mutant. Is it?

About the position of PCP-PPant-Glystab:

P10: "is its close proximity (3.6 Å) to the amino group of the Glystab moiety" Similar to comment above, 3.6 Å is too far for deprotonation, a shift of some kind needs to be evoked.

P6: "The overall orientation of the PCP domain relative to the C domain is similar to what has been observed in the structures of SrfA-C 12 and ObiF1 (PDB ID 6N8E)8 (SI Figure S2A-B)"

Are there crystal contacts involving the PCP domain, other than the PCP interacting with the acceptor site of the C domain? This is asked because there are some crystal contacts in the PCP domain of SrfA-C, and the packing in ObiF1 prevents its PCP from being able to assume the position seen in AB3403 and LgrA. It would be good to be able to state that PCP2 is only interacting with C2 at the acceptor site.

Figure 5: The glycine analog is missing its carbonyl group to make it will be stable. Would a bone fide PPant-glycine be able to assume the position observed? Figure 5b makes it appear like the carbonyl would clash with H2697.

P10: "It is important to note that Glystab sits in a different position to the aminoacyl mimic in a previous model of a C domain bound to the acceptor substrate – in these structures the aminoacyl mimic does not enter into the active site as far as observed in our GlyStab-PCP2-C3 complex.¹¹" Please elaborate with a more quantitative description, or preferably, a supplemental figure.

P10: "A significant energy barrier is observed for proton transfer from the zwitterionic intermediate to the imidazole group of the active site histidine residue, suggesting the mechanism of peptide bond formation in C domains relies on specific base catalysis. This may explain why the mutation of this central histidine residue does not completely abolish activity in some C domains, as an active site water molecule could instead play the role of an alternate specific base."

This passage seems confusing – how can there be multiple specific bases? Also, I believe the

suggestion that water can accept a proton in C domains is similar to the conclusion of reference 11, so that should be cited here.

Other:

Figure S11 / Figure 6: PPant ejection assays are notoriously difficult, so it is not surprising to see somewhat noisy mass spectra, and the deuterium shift is a welcome control. However, given the background in Figure S11, a more detailed description of how the quantitation that led to the percentages listed in Figure 6 is warranted, as is inclusion of the mass spectra for those other experiments in Figure 6.

(Note the second P is not capitalized in the title Ppant ejection)

Abstract – “we report the first structural snapshots”, “previously uncharacterized” - Most journals do not allow primacy claims

We are very grateful for the time and effort shown by these reviewers and appreciate their comments that we have incorporated into our revised manuscript. We have addressed all the points that were raised, and we believe that this has certainly improved our revised manuscript.

Reviewer #1 (Remarks to the Author):

The manuscript by Izore and colleagues describes the crystal structure of the condensation domain from the fuscachelin A biosynthetic protein FscG. The authors describe several structures of the PCP2-C3 didomain construct as well as the PCP3 domain. Importantly, the asymmetric unit of the didomain structures contains two independent molecules in which the PCP of one chain is docked in the acceptor site of the condensation domain of the other molecule. Thus, the structure illustrates the positioning of the holo-PCP in the acceptor site of the condensation domain. While other similar structures have been determined, the critical advance of this manuscript is that the authors have loaded the PCP with the amino acyl thioether analog of the pantetheine, thus demonstrating for the first time the structure of a holo PCP bound in the condensation domain acceptor site that is loaded with its amino acid substrate.

This allows the authors to make several observations. First, the authors use bioinformatic analysis to demonstrate that there is not an obvious “specificity pocket”, in contrast to what is seen with the NRPS adenylation domains. Additionally, the authors explore the role of several residues in the pantetheine pocket and the active site. Arg2577 is proposed to play a role in gating the pantetheine tunnel to control access to the active site. Glu2702, which replaces the canonical glycine of the C-domain HHxxxDG motif, is proposed to position the acceptor amino acid in the condensation domain for small amino acid substrates. The more common glycine is used with larger amino acids, where interactions with the substrate side chains can probably better position the α -amino group for the condensation reaction.

This paper is an important addition to our knowledge of the structural mechanisms of NRPS condensation domains. The structures are well determined and supported by functional and bioinformatic analysis. I have a few suggestions the authors may wish to consider to help clarify the writing and analysis in a few places.

Thank you very much for this positive review!

Pg 4. The sentence within the last paragraph of the introduction “The C domain is shown to be tolerant of a range of aliphatic acceptor amino acid acceptor substrates, with the limited

acceptance rationalized...” is a bit of a contradiction (tolerant/limited acceptance). Perhaps include the word “small” to read “...tolerant of a small range ...”. (Also note duplication of “acceptor”)

Both changes have been made as suggested.

Figure 2 and Pg 7, line 1. I tend to think of symmetry-related in crystals to refer to interactions between asymmetric units. Thus I do not think that molecule B should be highlighted as “symmetry” nor described as docked to a “symmetry-related” C3 domain (pg 7). The description of Pg 5, 3 lines from the bottom, is more accurate.

We have adjusted the figure and the text to refer to remove reference to symmetry-related molecules, and instead refer to these as the second chain in the asymmetric unit.

Figure 4. Apo and holo are generally used to describe a protein +/- a cofactor. Thus, using “apo” in panels A and B is not accurate, with “unloaded” being correct (as in the legend). Please also list contour level and any carve radius in the legend.

The figure has been adjusted as suggested, and the missing information included in the legend as requested.

Figure 5, panels A and B. Glu2702 is mislabeled as 2707. This is incorrect in the legend as well.

Thank you for spotting this mistake – we have corrected it in both the figure and the caption.

Pg 11. The authors should provide a few sentences to provide a brief description of the Spycatcher/Spytag system so that the reader does not need to go the reference to know whether this a short peptide, two large protein domains, or a biotin/streptavidin based system.

We have included a brief description of this system to provide the reader with the necessary background to this useful protein ligation technique.

Pg 13. “ zwitterionic”

Corrected.

Supplementary Material

A table of contents for the SI can help guide the reader.

We have added a table of contents for the SI as suggested.

Table S1. The Resolution entry for all four datasets is formatted as “Low Resolution limit

(High Resolution limit)”. While everyone would understand what is meant, the format is a bit unusual as in every other entry of the table, the entry not in parentheses is the data for the complete data set and the entry in parentheses is the highest resolution shell. Resolution data should simply be presented as “48.14 – 2.18” or if preferred “48.14 – 2.18 (2.3 – 2.18)”.

Updated as suggested.

Table S4. Please define ΔG

This is now included.

Figure S1. Please state in the legend whether the aligned halves are the N-terminal or C-terminal subdomains.

The aligned halves in this figure were the C-terminal halves; this has now been clarified in the legend.

Figures S8, S9, and S10, detailing the plausible mechanisms from the supplementary discussion are not present in the SI but rather are part of Figure 5 of the main text.

Thank you for picking up this oversight – we have corrected these errors in figure citation.

Please consider expanding the Legend to Figure S14, including a label for the X-axis. I understand the motivation of this figure and believe I understand how the information is presented but it took me a while to interpret this.

Thank you for this point; we have adjusted this figure to include an X-axis label and extended the legend.

Reviewer #2 (Remarks to the Author):

This is a paper that reports on the structure of an NRPS module with the aminoacyl group (stable analog) bound. This is one of the missing structures in the understanding of NRPS assemblies. It was also so good to see that PPant ejection still being used. I discovered it in 2007 and published it in 2008 before starting my own career so it brings back some wonderful memories.

It is an incredibly useful technique – thank you!

It was an enjoyable read. Some comments and questions. All are minor and even if the authors have no answer, the paper would still be appropriate for NComms.

Thank you very much – we appreciate your comments and insights.

Fig 1a the lighter colors and letters are hard to see for the structures in the NRPS assembly lines. I suggest making a) the width of the entire page. And move b) and c) around to accommodate.

We have adjusted the figure as suggested; we have also changed the colour scheme in Figure 6 along the same lines.

Figure four it would be nice to see an overlay of the apo vs acylated PPant to show how the configurations stayed the same and where they are dissimilar.

We have added this overlay in a new panel in Figure 4 as suggested.

This is a bit of an open-ended question but this could have a big impact if there was such a trend and it may be possible to answer these questions with this unique structure. Are there conserved residues in the C domain that bind PPant?, are there conserved residues in the C domain that define Acyl selectivity? I see that you did a pocket analysis. It is not entirely clear what this is and what the training set was but perhaps look for two details. 1) are there specific residues associated with the active site depending on the substrate that is loaded. 2) are there polarities of residues that define substrate specificities. The tricky part is that you have to half the active site in the donor and acceptor region and then ask these questions. It may be that this is currently not possible to identify if there is a true donor or acceptor region in the C domain with the currently available structures but if it is then it could be very interesting.

Thank you for prompting us to do this – we have explored the conservation of the PPant interacting residues (shown in SI Figure S5), which found similar trends to that seen with R2577, i.e. the conservation of these residues is broadly conserved depending on the stereochemistry selectivity shown by the C domains (LCL and DCL). This analysis has been added to the SI in SI Figure S11 and is also briefly discussed in the main text. To clarify the

analyses performed, we have added additional text in the manuscript concerning our analyses of the C domain pocket residues, which did not indicate any patterns of correlation. Due to the highly variable structure of the donor substrates and a lack of structural data concerning how these are accommodated within a C domain, we did not attempt to analyse the donor site as we did for the acceptor site.

“This forced us to explore alternatives to thioester-tethered amino acids, and we chose to use an analog of the aminoacyl-CoA with a thioether in place of the reactive thioester.” I see from the structure that the carbonyl is also missing, can the authors clarify what the actual non-hydrolyzable thioether is in this section? (I saw the structures are shown in the SI but would be nice to mention what the mimic is in this section.) Perhaps just change to the following “...a thioether, missing the reactive carbonyl, that makes the thioester susceptible to nucleophiles.”

We have adjusted the description here as suggested.

Regarding HHXXXDG vs HHXXXDE, is there a trend among substrates that are connected based on sequence gazing the understanding of the new structure and the type of amino acids or acids that are coupled? I know you tested the E to G mutation in vitro but in vivo do you think the condensations become more promiscuous? I.e. is this another gatekeeper for selectivity for condensation build into the system post A domain activation? I know the discussion has a description of selectivity but limited in the context of the condensation rxn.

Based on our analyses of this position and the different residues found to occupy it (SI Figure S16), we see that there is a shift towards smaller acceptor substrates when the G is replaced with another (larger) residue. We see from our in vitro studies that the replacement of the E with G does decrease activity in this specific case, and our computational substrate docking studies support the role of the E in stabilising the desired pose of the acceptor amine group. However, as the G to E mutation is relatively rare (it is more commonly A or H) we do not as of yet fully understand role of different mutants at this position.

Figure S11, I would not call it HRMS. As the resolution is quite poor compared to FT instruments and the isotopes for proteins are not detected. The mass accuracy is OK not amazing but works. Maybe just specify the resolution of the MS instead so you don't have to specify HRMS. The numbers of the masses have one too many sig figs for the PPant ejections. For example, 621.3177, found: 621.3049 this is 0.128 Da difference. This should be 621.32, found: 621.31 based on this mass accuracy. The deuterated experiment is a nice confirmation that eliminates the need for the super high mass accuracy.

We have adjusted the manuscript in line with these points and are very happy to hear that you agree with the use of the deuterated sample to aid in the clarity of these experiments.

The paper does not have a data availability section in the paper. Coordinates, MS data, etc.

Thank you for pointing out this oversight! We have added this section.

Of course, due to my personal bias I would also encourage to make the MS/MS spectra of all the synthetic compounds publicly available not just as copies in a PDF.

We have uploaded this data now to the ProteomeXchange Consortium via the PRIDE partner repository to make access to the data available.

Reviewer #3 (Remarks to the Author):

The paper by Izoré et al. presents a state-of-the-art multidisciplinary research of C-domain structure and selectivity in non-ribosomal peptide biosynthesis. Four novel crystallographic structures are reported with a resolution of ~2 Å. Expert analysis of these structures followed by biochemical characterization, site-directed mutagenesis, bioinformatic analysis and molecular modeling provided novel insights into the roles of key residues of the C-domain. Most notable is the new understanding of the role of the HHxxxDE pattern and R2577. A crystallographic structure of C-domain in complex with an acceptor PCP-domain bearing a substrate is reported for the first time, thus shaping further steps towards our understanding of structure-function relationship in NRPSs and related proteins.

The take-home message of the presented manuscript is clear. I believe, the new crystallographic structures enriched by extensive multidisciplinary analysis have the capacity to make an immediate contribution to mechanistic understanding of non-ribosomal peptide biosynthesis, though some parts of the work require clarification or revision.

My criticism is focused on the handling of the aspect of conformational plasticity of NRPSs and their complexes. As authors clearly state in the introduction "NRPSs are highly flexible and the interactions between individual domains change during the process of chain assembly". In this context, my major points are as follows:

1. First and foremost, if NRPSs and their complexes are known or expected to be flexible, then how was it even possible to characterize the reported PDB structures?

The confusion here seems to have arisen due to our poor choice of language in the sentence "NRPSs are highly flexible and the interactions between individual domains change during the process of chain assembly". We have now clarified this point and replaced this sentence with the following: "NRPS complexes are highly flexible, with domains connected by flexible linkers that allow the interactions between them to change during the process of chain assembly. However, the individual domains (and certain didomain complexes that represent meta-stable points along the catalytic pathway) are less dynamic and can be more readily studied by methods such as X-ray crystallography."

What novel approach was used in the work that made it possible to characterize such complexes?

While crystallography on complete NRPS complexes (i.e. with all NRPS modules) would be complicated by the flexible arrangement of the modular domains, by focusing on only the reasonably stable PCP-C₃ didomain construct we were able to crystallize and characterize this complex.

2. If NRPSs and their complexes are flexible, then I would expect a series of conformational variants to exist in equilibrium. Then, authors also continuously refer to their PDB models as

"structural snapshots". What conformational and functional states of the complexes are reflected in the reported PDB "snapshots" compared to the conformation ensemble of NRPSs and their complexes? Bioinformatic comparison of the obtained complexes with related PDB structures and also molecular modeling was reported, mostly in Supplementary material, but I did not immediately find the answer to my questions. Thus I believe this issue was not address sufficiently clear and should be presented, possibly, as a dedicated section. How flexible are the reported complexes? How representative these "static" models are in "dynamic" context of conformation ensemble of NRPSs and their complexes?

This confusion was likely caused in part due to our poor choice of words in the introduction. It has been shown for a number of different NRPS didomain complexes (A + PCP domains, Oxy + X domains, C + PCP domains) that these didomain pairings adopt highly similar structures in different crystal structures (and crystal forms), strongly supporting these as represent functionally relevant states. We indicate this in the main text for C + PCP complex and also show these in the supplementary information due to the limited number of figures we can show in the main article. While the dynamics of the didomain complex is certainly important for a better understanding of the catalytic cycle, this work focuses more closely on the structural determinants of substrate selectivity, which can be rationalised reasonably well from the static snapshots that represent these low-energy states.

Would they move/rearrange during MD simulation? The molecular modeling part left the impression that the PDB complexes were stable during MD. How does this conclusion coexists with the notion of NRPSs being highly flexible?

To clarify, we did not report MD simulation on the didomain complex and, therefore cannot comment on whether we would expect them to move/rearrange during MD simulation; we have only reported the MD data from simulations with the isolated C domain (i.e. without the PCP and Ppant) rather than the didomain structure. We clearly state that in the sentence: "Molecular dynamics simulations initiated from structures of the C₃ domain (with the PCP-PPant removed) highlight the intrinsically dynamic nature of the acceptor substrate channel and the important role that R2577 has in modulating its shape and size (SI Figure S5)." (first paragraph, pg 10).

While the stability of the didomain would certainly have an important role in the overall catalytic cycle of the pathway (especially the speed at which peptides could be produced) and warrants further investigation in future studies, in this paper we were focussed on using MD and molecular docking to look at (i) the intrinsic dynamics of the C-domain (e.g. side-chain dynamics) and how that influenced acceptor substrate selectivity and (ii), to identify residues in the C-domain that determined selectivity near the end of the Ppant-arm (i.e. where the substrate is attached), respectively. These computational results support and helped to guide experimental work that focused on substrate selectivity, rather than domain-domain stability or dynamics. We considered running MD simulation of the pseudo-didomain complex to investigate the stability of the didomain complex. However, such dynamics are likely to occur on a much longer time scale than the sidechain motions we were interested in and would

require much longer MD simulations (or enhanced sampling methods). Since interdomain dynamics was not a focus of the current study, we did not feel it necessary to pursue these investigations in this work.

3. Authors report molecular docking and transition state calculations. These are well-known and widely used approaches, but they all demand a single "static" high-quality model as input. In practice, this usually means that the protein in question should be relatively "rigid". I had an impression that authors used the original "static" PDB models as input to these methods. Since NRPSs and their complexes are considered highly flexible, I wonder about the validity of such approach. Either there is a justification for use of "static" models of the complexes in question for docking and QM, or additional work should be performed, e.g. docking into several structural states and snapshots obtained from PDB or molecular dynamics.

Molecular Docking. The reviewer is correct that the substrate-docking calculations were performed based on the static PDB models of the C-domain (obtained from the 1.90 Å Gly_{stab}-loaded crystal structure structure). As the review mentions, the use of a rigid receptor is a "well-known and widely used" approach for trying to identify key residues involved in substrate binding (i.e. to be used to guide or support experimental data), as we were doing here. We do not believe it is necessary to perform substrate-docking calculations using an ensemble-based approach for several reasons:

- 1. **Regarding confusion about NRPS flexibility:** Firstly, we emphasise that our sentence regarding NRPSs being flexible was trying to illustrate that the individual modules/domains rearrange relative to each other (slow time scales, likely ~seconds) rather than referring to the intrinsic flexibility of the C-domain, or the short-term stability of the C domain-PCP complexes. The purpose of the substrate-docking studies was to dock alternate substrates into the low-energy state of the C domain represented by the crystal structures. Considering each of the crystal structures showed similar complexes (in terms of relative positioning of PCP/C domain and the docked position of PPant), we are confident they represent a functionally-relevant low-energy state and that these crystal structures were suitable for our substrate-docking studies.*
- 2. **Regarding the dynamics of the C domain:** As the reviewer correctly mentions, molecular docking "demands a single "static" high-quality model as input". We believe that the structure of the C domain used for molecular docking was both high-quality (1.90 Å) and rigid enough to not require ensemble-based approaches. Certainly, our MD simulation initiated from the C domain structure did highlight fast protein motions (ps – ns timescales), including side chain motions (**Figure S5 a,b**) and small amounts of "breathing" between the two halves of the C-domain (**Figure S5 c,d**). While we acknowledge these motions lead to an ensemble of receptor conformations that could potentially have been used to dock substrates (i.e. by docking substrates into several snapshots obtained from MD), we are confident that this would have not lead to different residues being identified as key*

substrate-binding residues. Again, we emphasise that substrate-docking studies were conducted to guide and support the experimental results only; had we been interested in calculating relative binding affinities from these calculations, an ensemble-based approach (as the reviewer suggested) may have been appropriate.

- 3. **Regarding similarities between docked substrates:** the alternate substrates only vary at the termini where the substrate is attached to the Ppant-arm, and we therefore felt it was reasonable to use the assumption that the Ppant arm would bind in the tunnel in a similar manner in each case, similar to what we observed in the Gly_{stab} structure (and other structures). Since we were largely interested in identifying potential interactions between the PPant bound substrate and the active site residues, we believe it is reasonable to perform docking using the static crystal structure of the C domain (from the Gly_{stab} structure). Certainly, the active site cavity is large enough to allow significant side-chain motions of the active site residues (also observed in MD data) – however, we emphasise that molecular docking was only used to identify potential residue interactions (and to guide and support the more important experimental data), rather than to rank binding affinities of the substrates etc (which would require a more thorough approach, including the use of induced-fit approaches or ensemble docking, as suggested by the reviewer).*
- 4. **Ligand flexibility:** With regards to the flexibility of the Ppant-substrate substrates: these were allowed to be completely flexible during the docking algorithm. In addition, only a single “soft” restraint was used to hold the phosphate motif of the Ppant arm in a position near to what was observed in the crystal structures of the protein (as described in the methods section). This allows some movement of the Ppant arm relative to the C₃ domain.*

Transition state calculations. *The transition state calculations involve a small model reaction taking place in solution. They do not attempt to model the exact binding orientation of the substrates within the enzyme active site. The purpose of these calculations was simply to determine whether the amide bond-forming reaction has an intrinsic preference for a stepwise or concerted mechanism.*

To clarify this, we have expanded the text on p. 10 to read as follows:

“In order to determine whether the intrinsic mechanistic preference of the amide bond-forming reaction is stepwise or concerted, we calculated the reaction of a model donor, acceptor, and imidazole base in solution with density functional theory (Figure 5C, see Supplementary Discussion for details of the mechanistic investigation). The attack of the model amine on the thioester strongly prefers a stepwise mechanism in which N-C bond formation precedes N deprotonation by the imidazole, rather than a concerted mechanism in which these two events take place simultaneously. Therefore we predict that the enzyme-catalyzed amide bond formation likely involves a similar sequence, with a distinct zwitterionic (oxyanion/ammonium) intermediate (Figure 5D).”

Other issues:

1. Regarding the PDB structures

The four crystallographic models present primary result of the work, and yet the respective PDB codes were not mentioned even once in the main text. Instead, the codes and explanation of their contents are buried deep in the Supplementary data. I would like to see the PDB codes and a brief explanation of contents of the respective 3D-structures in a dedicated section of the main text. The figures obtained using the original models should also cite their codes. I even suggest to additionally provide a list of the codes in the abstract, for convenience of the users.

We have now added the PDB code references for our structures in the main text. As the structures are all introduced during the manuscript, we feel an additional paragraph listing this is not required.

Then, unless there is some specific rule against it that I am not aware of, I would like to see all four PDB structures (i.e. files which I can load into PyMol and view in 3D) before submitting my final decision. In recent years, there is a growing amount of PDB entries containing obvious human-visible errors. The formal validation reports are interesting, but without the PDB models themselves they do not show the full picture. So I believe an extra check of PDB files by independent pair of eyes will be useful to both the authors and further users of this data.

We have happily included these files in the revision for the reviewers to access if they so desire.

2. Bioinformatic analysis

A multiple sequence alignment tool was used to superimpose sequences of individual domains for bioinformatic analysis. Sequence alignment is limited to relatively close homologs with a high level of pairwise sequence similarity (at least ~30-40% is needed to hope for a meaningful comparison, as everything below 40% is commonly considered as the "twilight zone" of sequence alignment). However, the level of sequence identity between the aligned sequences was not stated in Supplementary material (Bioinformatic methods section). The level of pairwise sequence similarity should be stated. If it resides below 40%, other methods should probably be used (e.g. 3D-alignment, profile-based alignment, structure-guided sequence alignment, etc.), or additional justification should be provided.

C domains have been shown to be highly conserved homologs (Rausch et al, 2005 doi: 10.1093/nar/gki885 & Ziemert et al, 2012 doi: 10.1371/journal.pone.0034064), so the MSAs are valid. We nevertheless calculated the average distances of the conserved areas of the C domains to be 49.4%.

Authors discuss sequence conservation statistics in the manuscript (e.g. 80% of Gly is found in DCL domains in positions equivalent to R2577, etc.). It usually makes sense to calculate such statistics after the removal of redundant, duplicated, incomplete, etc. sequences (i.e. the redundancy filter). No such filter was stated in methods, as it seems 1456 LCL and 593 DCL sequences were collected as is, and may contain redundant information. The same concern also applies to the calculation of correlation coefficients.

We thank the reviewer for this comment. Indeed, in analyses of random sequences from a database the removal of redundant sequences would be suitable. However, because we used the highly curated MiBiG database of experimentally verified sequences, which is non redundant, this filtering step is not necessary.

3. Molecular modelling

I am concerned by the Molecular dynamics protocol. An orthorhombic box of 10 Å around the protein molecule was used for simulation of a potentially flexible protein-protein complex. Such a small gap between the edge and the box means that anticipated relative movement of stacked subunits during the simulation would cause them to interact with their images in neighboring cells (assuming that periodic boundary conditions were used, which, I think, was not clearly stated, but implied by the protocol). One of many potential side-effects of such small-box setup could be artificial over-stabilization of the complex. If indeed the authors expect their complexes to implement intrinsic conformational flexibility, a box with 30Å or so should probably be used instead. Even such a huge box would not be sufficient if subunits move too much, but in the absence of significant major rearrangements in the protein-protein complexes such box would do to present a valid simulation, ruling out artificial over-stabilization. Authors should clarify their simulation protocol and the choice of this crucial parameter, or perform additional simulations.

The reviewer raises concerns about the box size used during our simulations of the C-domain. We agree that the choice of box size is an important consideration when performing MD. However, we feel that the box size used (10 Å buffer around the protein) for the simulations can be justified:

- 1. **Simulations were only performed on the isolated C₃ domain, not the didomain complex.** The reviewer was concerned that the box size was not appropriate for simulating the di-domain complex. However, for this study we only simulated the isolated C₃ domain, as is clearly stated in the top paragraph on page 10: “Molecular dynamics simulations initiated from structures of the **C₃ domain (with the PCP-PPant removed)** highlight the intrinsically dynamic nature of the acceptor substrate channel and the important role that R2577 has in modulating its shape and size (SI Figure S5).” We have also made slight changes to the figure legend for Figure S5 to make this clear here as well. We agree that a larger box size would have likely been necessary for simulation of a protein-protein*

complex, since larger motions would have been expected during the simulation. However, because we only simulated the C₃ domain, and the overall size/shape of this (near-spherical) domain did not change significantly during our simulations (See Radius of Gyration analysis below), we feel the box sized used is appropriate.

2. **Standard box size in Desmond:** the box size used is the default box size recommended in the Desmond Molecular Dynamics wizard and an orthorhombic box with 10 Å buffer around the protein has been used for numerous other published MD studies [including refs. 1–4]. The 10 Å buffer is provides at least 20 Å between any two neighbouring protein molecules when periodic boundary conditions are applied; even if the protein was to increase in size, or tumble, during the simulation, this allows a significant buffer between neighbouring molecules. In particular, the 20 Å separation is significantly larger than the 9 Å distance used for the electrostatic cut-off, and we therefore feel that this would avoid any of the artefacts the reviewer is concerned about, while helping to minimize the size of the MD system and the computational time required.
3. **We were primarily focussed on side-chain motions of internal residues:** the primary reason for performing the MD simulations of the isolated C₃ domain was to investigate the sampling of internal sidechains, especially that of the Arg residue in the substrate tunnel. We believe that the sidechain sampling would be unlikely to be significantly influenced by any (small amount of) artificial stabilisation of the protein due to a smaller box size.
4. **MD was only provided to support experimental data:** The data obtained from our MD simulations did not make up the central argument for this paper, and we feel it unnecessary at this point to repeat the MD simulations with a larger box size (which would require computational time, and likely lead to minimal changes in our interpretation of the results regarding side-chain dynamics).

As suggested by the reviewer, we have updated the methods section to clarify the use of PBCs, including some justification of the choice of box size. We thank the reviewer for this suggestion.:

“...then placed in an orthorhombic box with a buffer of 10 Å around the protein molecule, and periodic boundary conditions (PBCs) were applied. This provided sufficient distance between neighboring protein molecules once PBCs were applied (~20 Å); this distance was significantly larger than the 9 Å electrostatic cut-off used during simulations.”

Supporting Figure (for reviewers only). Radius of Gyration (ROG) analysis of the six MD simulations presented in this study. During the simulation of the isolated C_3 domains (both from the unloaded and Gly_{stab}-loaded structures), the overall ROG only increased a small amount (no more than 0.5 Å).

1. Bertoldo JB, Rodrigues T, Dunsmore L, Aprile FA, Marques MC, Rosado LA, Boutoureira O, Steinbrecher TB, Sherman W, Corzana F, et al.: **A water-bridged cysteine-cysteine redox regulation mechanism in bacterial protein tyrosine phosphatases.** *Chem* 2017, **3**:665–677.
2. Dagbay KB, Bolik-Coulon N, Savinov SN, Hardy JA: **Caspase-6 undergoes a distinct helix-strand interconversion upon substrate binding.** *J Biol Chem* 2017, **292**:4885–4897.
3. Guo S, Xu J, Pavlidis IV, Lan D, Bornscheuer UT, Liu J, Wang Y: **Structure of product-bound SMG1 lipase: active site gating implications.** *FEBS J* 2015, **282**:4538–4547.
4. Zhu W, Radadiya A, Bisson C, Wenzel S, Nordin BE, Martínez-Márquez F, Imasaki T, Sedelnikova SE, Coricello A, Baumann P, et al.: **High-resolution crystal structure of human asparagine synthetase enables analysis of inhibitor binding and selectivity.** *Commun Biol* 2019, **2**:345.

I wonder what was the rationale for selecting OPLS3e force field for modeling, given the two key aspects of the study: (1) the need to model the expected conformational plasticity and (2) the need to model crucial interactions involving charged residues (R2577 in particular)?

We agree with the reviewer that the choice of force fields is an important aspect of MD simulation design and that our choice of OPLS3e would benefit from justification as to its choice. As the reviewer highlights, the key aspects of the MD study was to 1) model conformational plasticity (i.e. standard protein dynamics) and modelling charged residues. Firstly, OPLS3e is the default force field used in Desmond and is based on the OPLS3 force field [5]. It is an all-atomistic force field, and we believe represents a good all-round force field for a range of simulations. While the updated OPLS3 version documentation does

emphasise improvements such as the extensive parameterization of drug-like molecules, it also has significant improvements in the protein force field and the charge model. Further, it has optimised parameters for interaction with substrates, which is relevant to this study. It has been used extensively for the simulation of protein and protein-substrate interactions (including interactions in which charged interactions play an important role) including recent papers (e.g. [6–8]). Further, a recent article shows that OPLS3e outperforms other common force-fields, for reproducing geometries and conformational energies compared with quantum mechanical data [9].

We have avoided using a force field that is specialised for very specific applications (e.g. the simulation of lipids), instead choosing OPLS3e because it represents a good all-round force field for simulating protein motions.

Again, even if the reviewer could argue a more suitable force-field, we believe that OPLS3e is more than adequate for these simulations, especially considering that the MD simulations were not the central focus of this work.

As suggested by the reviewer, we have updated the methods section to provide some justification for our choice of force field. If the reviewer remains concerned, we please ask that they provide a more detailed reason, including their suggestion for FF choice for this type of simulation.

The methods section now reads: “The OPLS3e force field³⁰ was used at all stages of the simulation. The OPLS3e force field is the default force field in Desmond and performs well against other force fields for the simulation of protein molecules.³⁰”

5. Roos K, Wu C, Damm W, Reboul M, Stevenson JM, Lu C, Dahlgren MK, Mondal S, Chen W, Wang L, et al.: **OPLS3e: Extending Force Field Coverage for Drug-Like Small Molecules**. *J Chem Theory Comput* 2019, **15**:1863–1874.

6. Yoshino R, Yasuo N, Sekijima M: **Identification of key interactions between SARS-CoV-2 main protease and inhibitor drug candidates**. *Sci Rep* 2020, **10**:12493.

Treatment of Internal and Surface Residues in Empirical pKa Predictions. *J Chem Theory Comput* 2011, **7**:525–537.

7. Lane JR, Abramyan AM, Adhikari P, Keen AC, Lee K-H, Sanchez J, Verma RK, Lim HD, Yano H, Javitch JA, et al.: **Distinct inactive conformations of the dopamine D2 and D3 receptors correspond to different extents of inverse agonism**. *Elife* 2020, **9**.

8. Kaczmarek JA, Mahawaththa MC, Feintuch A, Clifton BE, Adams LA, Goldfarb D, Otting G, Jackson CJ: **Altered conformational sampling along an evolutionary trajectory changes the catalytic activity of an enzyme**. *Nat Commun* 2020, **11**:5945.

9. Lim VT, Hahn DF, Tresadern G, Bayly CI, Mobley DL: **Benchmark assessment of molecular geometries and energies from small molecule force fields**. *F1000Res* 2020, **9**:1390.

ProPKA version should be stated, as different releases of this particular software are known to give significantly different results.

We thank the reviewer for raising this concern – we were not aware that different versions of ProPKA could lead to varied results. We used the ProPKA version that was part of the

Schrodinger 2019-1 release, which was ProPKA 3.[10,11] We have updated the methods section to include the ProPKA version.

*10. Olsson MHM, Søndergaard CR, Rostkowski M, Jensen JH: **PROPKA3: Consistent Treatment of Internal and Surface Residues in Empirical pKa Predictions.** J Chem Theory Comput 2011, 7:525–537.*

*11. Søndergaard CR, Olsson MHM, Rostkowski M, Jensen JH: **Improved Treatment of Ligands and Coupling Effects in Empirical Calculation and Rationalization of pKa Values.** J Chem Theory Comput 2011, 7:2284–2295.*

The "Results" section of the main text mentions the term "docked" / "docking" multiple times. The authors should clarify the meaning of the term, as it was not immediately clear to me. Does it refer to a predictive computational method called "molecular docking", or does it refer to a "stacking" of protein subunits observed in crystallographic complexes obtained experimentally.

We thank the reviewer for pointing this confusion out. We have tried to clarify this in the text by removing instances of “docked” when talking about the position of the PCP or PPant arm in the crystal structures of the complexes (instead using “positioned” etc). When discussing computational docking of ligands or computational docking of the PCP onto C3, we have used terms such as “computational docking”. We have changed this in the main manuscript, the supplementary information and methods sections. We hope this clarifies the manuscript.

There is a section called "Supplementary Discussion" discussing a computational investigation of the catalytic mechanism. I was neither able to quickly find a reference to this section in the main text, nor the QM calculations were discussed in the Methods. Thus, I was not able to immediately understand the value of this supplementary section, its results and relevance to the study.

We have now included a description of the DFT calculations in the Methods and inserted a reference to the supplementary discussion on p.10 of the manuscript.

Reviewer #4 (Remarks to the Author):

In “Understanding condensation domain selectivity in non-ribosomal peptide biosynthesis: structural characterization of the acceptor bound state”, Cryle and colleagues present a nice series of structures of a PCP-C didomain, in which the PCP domain is docked at the acceptor site of the adjacent C domain. Structures in which the PCP is loaded with apo PPant show this moiety to curl away from the active site, but when a nearby R2577G is mutated, or when a propylamine representing glycine is attached, the PPant enters the active site.

The authors set up a useful PCP2-C3-PCP3 system with Spycatcher to specifically load PCP2 and PCP3 with different PPant moieties and use it to present Gly, Ala, Leu or Phe to the C domain, and to analyze three C domain mutations.

Overall, the manuscript is well written and scientifically sound. It is not clear from the data presented whether some specific points, such as the putative gating role for R2577, are a general feature of NRPS biology, but this paper will be a nice addition to the NRPS literature.

Thank you very much for your time and helpful comments!

Specific comments and questions:

About R2577:

SI Figure S6: R2577 is conserved in 70% of LCL domains. Can the authors comment on why they think the next most common residues are G, A, Q and S? Also, what is the conservation in other C-type domains, for example Cyc domains or starter C domains? It seems odd that this is not conserved in DCL domains given that the acceptor substrate has L chirality.

We do not yet fully understand the significance of different potential mutations at this position; however, we do also now see that the PPant interacting residues also seem to follow the same trend. We performed an analysis of the R2577 position in starter C domains (152) as suggested and have incorporated this into SI Figure S6); this shows a much wider range of amino acids are found in this position in such C domains. For other C domains, such as heterocyclization domains, the number of examples of these is too low (6) to allow meaningful analysis of this position. In terms of the reasons why R is conserved in LCL yet not in DCL domains, this remains unresolved, although the variation in conservation between D and L configured donor substrates does suggest there is a link between machineries that provide a D-configured donor and those that do not need to epimerize their donor substrates.

P8: “ LCL* ” What does the asterisk signify here?

Apologies for the confusion – this asterisk refers to the footnote where this nomenclature is described.

P10: “R2577 now forms specific interactions with two of the carbonyl oxygen atoms in the Ppant arm (3.7 Å and 3.8 Å)”

These are long distances and would be quite weak interactions.

We have altered the phrase from “specific interactions” to “weak interactions” as suggested.

P13: “One hypothesis for the role of this residue would be to prevent the unwanted “pass-through” of donor substrates without elongation (e.g. from PCP2 to PCP3).” Is pass-through a likely event, given that the nucleophile in the pass-through reaction (PPE thiol) is 3 atoms away from the nucleophile peptide bond formation (amino group). If this were an important mechanism, pass through should be observable in the R2577G mutant. Is it?

This is an excellent point, although unfortunately it is a very challenging experiment to perform – it requires us to be able to distinguish the loaded state of either the donor PCP and acceptor PCP within the same protein construct (i.e. the linked Spyttag/Spycatcher PCP-C-PCP construct). PPant ejection would result in the same fragment in either case, and the overall mass of the protein would not change. The only route to explore this further would require the chemoenzymatic synthesis of specifically labelled coenzyme A, which is highly challenging and has not yet been accomplished in our laboratory.

About the position of PCP-PPant-Glystab:

P10: “is its close proximity (3.6 Å) to the amino group of the Glystab moiety” Similar to comment above, 3.6 Å is too far for deprotonation, a shift of some kind needs to be evoked.

We agree - as the His isn't close enough to deprotonate the amino group as it attacks the carbonyl group. However, once the tetrahedral intermediate is formed the His is close enough to the amino group for proton transfer to occur, as our calculations show that C-N bond formation alters the position of the amino group relative to the His.

P6: “The overall orientation of the PCP domain relative to the C domain is similar to what has been observed in the structures of SrfA-C 12 and ObiF1 (PDB ID 6N8E)8 (SI Figure S2A-B)”

Are there crystal contacts involving the PCP domain, other than the PCP interacting with the acceptor site of the C domain? This is asked because there are some crystal contacts in the PCP domain of SrfA-C, and the packing in ObiF1 prevents its PCP from being able to assume the position seen in AB3403 and LgrA. It would be good to be able to state that PCP2 is only interacting with C2 at the acceptor site.

The reviewer's question concerns the three C domain-PCP complex structures we have reported in this manuscript, PDB 7KVV, 7KW0 and 7KW2. Each of these crystallized in the same space group ($P2_12_12_1$) and are packed in the crystal lattice in a similar manner, with

two C_3 -PCP chains within each asymmetric unit. As described in the manuscript, each of these PCP domains interact at the acceptor site of a C_3 domain in a neighboring asymmetric unit – this is what we have referred to as the pseudo-acceptor C_3 -PCP complex.

In addition to these contacts, there is a small contact ($\sim 186 \text{ \AA}^2$ interface area as determined by PDBePISA) that occurs between each PCP domain and the neighbouring C_3 domain within the same asymmetric unit in each crystal structure (i.e. that are not part of the C_3 -PCP pseudoacceptor complex, figure below).

While this is less substantial than the crystal packing interfaces observed in the tightly-packed SrfA-C structure (PDB 2VSQ), for example, we acknowledge that the presence of this crystal packing interface may influence the orientation of the PCP relative to the “acceptor” C_3 domain. Having said that, we note that the computational molecular docking of PCP₃ onto the C_3 domain arrives at an almost identical position in terms of binding interface. Furthermore, any alteration of PCP binding (i.e. to match the orientation of those seen in AB3403 and LgrA) still places PCP’s modified serine in a similar position, meaning that the PPant will sit at the same location relative to the C_3 domain channel and active site (the focus on this study). Therefore, we feel that any minor artifacts in the PCP-orientation that may be caused by crystal packing will not impact on the primary conclusions of this work.

Supporting Figure (for reviewers only). Crystal contacts involving the PCP-domain. Shown here is the interface between one PCP-domain and the neighbouring C-domain within the same asymmetric unit (from the WT PCP₂-C₃ PPant structure). Similar packing interactions are observed for the other PCP domain within the asymmetric unit, as well as in the other two PCP-C₃ complex structures reported in this work.

Figure 5: The glycine analog is missing its carbonyl group to make it will be stable. Would a

bone fide PPant-glycine be able to assume the position observed? Figure 5b makes it appear like the carbonyl would clash with H2697.

While we do not have a crystal structure of with PPant-glycine, we have performed computational docking of the “bone fide” PPant-glycine into the structure of the C3 domain; this provides some insight into how the PPant-glycine would likely dock in the C3 domain (SI Figure S14). In particular, the reviewer was concerned that the carbonyl group would clash with H2697. The top poses obtained during computational docking of the alternate substrates (including PPant-Gly) does highlight some flexibility in the position of carbonyl group of the PPant as can be seen in Figure S14; sometimes the carbonyl points towards His2697, but in other poses the carbonyl points towards the opposite side of the tunnel (e.g. towards Met2917). We therefore believe the bone-fide PPant-glycine would bind in a very similar way to Glystab-PPant, but that it is possible that the PPant arm would bind in a slightly different orientation to allow for additional room for the carbonyl group if needed.

P10: “It is important to note that Glystab sits in a different position to the aminoacyl mimic in a previous model of a C domain bound to the acceptor substrate – in these structures the aminoacyl mimic does not enter into the active site as far as observed in our GlyStab-PCP2-C3 complex.11” Please elaborate with a more quantitative description, or preferably, a supplemental figure.

This is an excellent suggestion, and we have added a comparison of these two structures as a new figure in the supplementary information (SI Figure S12).

P10: “A significant energy barrier is observed for proton transfer from the zwitterionic intermediate to the imidazole group of the active site histidine residue, suggesting the mechanism of peptide bond formation in C domains relies on specific base catalysis. This may explain why the mutation of this central histidine residue does not completely abolish activity in some C domains, as an active site water molecule could instead play the role of an alternate specific base.”

This passage seems confusing – how can there be multiple specific bases? Also, I believe the suggestion that water can accept a proton in C domains is similar to the conclusion of reference 11, so that should be cited here.

We agree that this was confusingly written – the nomenclature of specific vs general base catalysis follows classical physical organic chemistry nomenclature and we have re-written the relevant parts of the manuscript to concentrate simply on the results of our studies (in terms of C domain mechanism) rather than descriptions using the specific/ general terminology. The reference has been added as indicated, thanks for pointing this out!

Other:

Figure S11 / Figure 6: PPant ejection assays are notoriously difficult, so it is not surprising to

see somewhat noisy mass spectra, and the deuterium shift is a welcome control. However, given the background in Figure S11, a more detailed description of how the quantitation that led to the percentages listed in Figure 6 is warranted, as is inclusion of the mass spectra for those other experiments in Figure 6. (Note the second P is not capitalized in the title Ppant ejection)

The experiments we used to quantify the conversion of tripeptide into tetrapeptide shown in Figure 6 were not performed using PPant ejection. Instead, the analysis of these reactions was performed using HRMS/MS2 on samples that had been cleaved from the PCP domains by the addition of methylamine. Examples of the analyses for each of these reactions is shown in SI Figure S28-S33, which we now have added references to in the caption of Figure 6 and also the main text. Furthermore, we have now uploaded all HRMS data to the ProteomeXchange Consortium via the PRIDE partner repository to make these generally available. We also fixed the error in the title of SI Figure S11 – thanks for spotting it!

Abstract – “we report the first structural snapshots”, “previously uncharacterized” - Most journals do not allow primacy claims

Thanks for pointing out our oversight – we’ve modified the abstract to remove these.

REVIEWERS' COMMENTS

Reviewer #1 (Remarks to the Author):

The authors have addressed my prior concerns and I believe this is suitable for publications. Congratulations on the great structures and accompanying analysis.

Andrew Gulick

Reviewer #2 (Remarks to the Author):

All my comments are addressed. Nice job and congrats on the paper.

Reviewer #3 (Remarks to the Author):

The authors responded to my comments with great attention and in precise detail. I am now pleased to recommend publishing the revised manuscript in its present form.

Dmitry Suplatov
Lomonosov Moscow State University

Reviewer #4 (Remarks to the Author):

The manuscript can be published.